# Adaptive Clipping for Differential Private Federated Learning in Interpolation Regimes

**Takumi Fukami** *takumi.fukami@ntt.com*
*NTT Social Information Laboratories*

**Tomoya Murata** *murata@msi.co.jp*
*NTT DATA Mathematical Systems Inc.*

**Kenta Niwa** *kenta.niwa@ntt.com*
*NTT Communication Science Laboratories*

**Reviewed on OpenReview:** *https://openreview.net/forum?id=vvSHlH3a8V*

## Abstract

We investigate improving the utility of standard differential private optimization algorithms by adaptively determining the clipping radius in federated learning. Our adaptive clipping radius is based on the root-mean-square of the gradient norms, motivated by the interpolation property and smoothness of the objectives. In addition to Renyi Differential Privacy (RDP) analysis, we conduct theoretical utility analysis of the proposed algorithm, showing that our method enhances utility compared to DP-SGD for smooth and non-strongly convex objectives. Numerical experiments confirm the superiority of our adaptive clipping algorithm over standard DP optimization with fixed clipping radius in federated learning settings.

## 1 Introduction

Federated Learning (FL) is an emerging distributed learning paradigm in machine learning. In this framework, each client possesses a local dataset, and these clients work collaboratively to train a shared machine learning model by exchanging information through communication (Konečnỳ et al., 2016; McMahan et al., 2017). To preserve the privacy of client information, federated learning prohibits direct sharing of local datasets. Instead of sharing the raw datasets, federated learning algorithms try to learn a model only by exchanging the local model parameters or the gradients computed on the local datasets. However, this approach does not fully protect the privacy of the local datasets. In fact, it is possible to extract private information from the communicated local models or gradients. Major attacks designed to steal private information include membership inference attack (Shokri et al., 2017; Yeom et al., 2018; Nasr et al., 2019) and reconstruction attack (Fredrikson et al., 2015; Zhu et al., 2019; Yang et al., 2019; Wang et al., 2019b; Geiping et al., 2020; Zhang et al., 2020).

Differential Privacy (DP) has recently received much attention to protect client data privacy against the aforementioned attacks. DP provides a framework that measures the extent to which privacy information leaks when using a given random mechanism (Dwork et al., 2006; Dwork & Naor, 2010; Dwork et al., 2010; Dwork, 2011; Dwork et al., 2014). To ensure the DP guarantees of a federated learning algorithm, existing works typically rely on the gradient perturbation technique(Song et al., 2013; Abadi et al., 2016; McMahan et al., 2018; Geyer et al., 2017; Triastcyn & Faltings, 2019). For example, Differential Private Stochastic Gradient Descent (DP-SGD) is a workhorse of DP optimization. In DP-SGD, independent Gaussian noise with a mean of zero is added to the computed gradient, resulting in a noisy gradient that is then used to update the parameters at each iteration. The algorithm determines the standard deviation of this noise, referred to as the DP-noise size, based on the desired privacy level.

In DP optimization, a fundamental trade-off exists between the optimization accuracy (often referred to as utility) and the privacy level. Thus, it is crucial to compare the utility of the algorithms at a given privacy level in order to evaluate the performance of the DP optimization algorithms. For instance, DP-SGD is known to achieve a utility of $O(\sqrt{d\log(1/\delta)}/(n\varepsilon))$ with a central $(\varepsilon, \delta)$-DP guarantee, where $d$ represents the problem dimension and $n$ is the total number of samples. This rate is indeed minimax optimal in non-strongly convex settings (Bassily et al., 2014). Given certain privacy levels, the development of a DP optimization algorithm with improved utility attracts much interest in the field due to its fundamental importance.

To enhance utility, a promising approach is to use adaptive clipping (Pichapati et al., 2019; Asi et al., 2021; Andrew et al., 2021; Yang et al., 2022; Bu et al., 2022; Xia et al., 2023). Standard DP optimization algorithms often use gradient clipping to bound the $L2$-sensitivity of communicated parameters, with a fixed clipping radius (Abadi et al., 2016). In contrast, adaptive clipping dynamically adjusts the clipping radius based on observed gradient values, leading to improved utility. In particular, iteration-level adaptive clipping aims to reduce the DP noise size by estimating the iteration-dependent $L2$ sensitivity from the gradient information. A key implicit assumption underlying the iteration-level adaptive clipping is the interpolation property of the objectives, meaning that per-sample gradient norms converge to zero as the model parameter approaches the optimum. This property is often observed in high-dimensional problems, such as training deep neural networks. With the growing prevalence of large-scale machine learning models, the interpolation property has become much more realistic, further highlighting the importance of iteration-level adaptive clipping.

However, many iteration-level adaptive clipping algorithms are primarily heuristic, lacking rigorous utility analysis and theoretical improvements over non-adaptive clipping algorithms. While empirical effectiveness has been reported, a solid theoretical foundation is often missing. The objective of this study is to develop a novel iteration-level adaptive clipping algorithm for federated learning and provide a theoretical utility analysis. Additionally, we conduct numerical experiments to demonstrate the superiority of the proposed method over non-adaptive clipping approaches.

**Main Contributions**

We propose a new adaptive clipping algorithm for *local record-level* DP federated learning, which is particularly relevant in cross-silo FL. The proposed per-round adaptive clipping radius $C_r$ is determined based on the root-mean-square gradient norm $\sqrt{(\tau/n)\sum_{z\in D}\|\nabla\ell_z(x_{r-1})\|^2}$ at the current global model $x_{r-1}$, which becomes smaller as the optimization proceeds, and reduces the amount of DP noise added to the model parameter in local optimization by exploiting the interpolation property of the objective function, where $\tau$ is the predefined parameter controlling the clipping bias.

We provide not only DP guarantees of the algorithm based on RDP technique (Mironov, 2017), but also a utility analysis. In our utility analysis, the main technical difficulty is to show the utility improvement over DP-SGD by carefully quantifying the trade-off between the reduction of the adaptive clipping radius $C_r$ (and the DP noise size) and the clipping bias depending on the choice of $\tau$. Furthermore, we analyze the impact of local optimization bias and the stochastic gradient noise, as our algorithm is based on FedAvg.

For smooth convex objectives under interpolation regimes, we show that our algorithm with $(\varepsilon, \delta)$-DP guarantees achieves a utility of $\widetilde{o}(\epsilon_{\text{util}}^{\text{DP-SGD}})$ with a factor depending on $\tau$ and $\tau_q$, where $\epsilon_{\text{util}}^{\text{DP-SGD}} := \widetilde{\Theta}(\sqrt{d}/(n_{\text{loc}}\sqrt{P}\varepsilon))$ is the best-known utility of DP-SGD under local record-level DP settings, and $\tau_q$ is the mean-quantile ratio as defined in Definition 1. Our result indicates a theoretical utility improvement over the best-known one when the mean-quantile ratio $\tau_q$ is not too large, which depends on the model and dataset. To validate this finding, we conduct numerical experiments to demonstrate empirical superiority over DP-FedAvg, which is a natural counterpart in federated learning of DP-SGD.

**Related Works**

Here, we briefly discuss the relationship between existing studies on adaptive clipping and this study.

One primary motivation for adaptive clipping is to enhance utility compared to fixed clipping methods such as DP-SGD and DP-FedAvg. Fu et al. (2022) studied an iteration-level adaptive clipping method for

DP federated learning, utilizing the mean of the (non-squared) gradient norm with heuristic DP noise size scheduling. Despite its empirical efficacy, this approach lacks utility analysis and fails to achieve theoretical improvements over the non-adaptive clipping methods. In contrast, we delve into the theoretical utility of the proposed algorithm and demonstrate both empirical and theoretical enhancements over the non-adaptive clipping methods. Pichapati et al. (2019) introduced a coordinate-level adaptive clipping method based on gradient normalization and de-normalization techniques. Asi et al. (2021) proposed PAGAN, which is a combination of a coordinate-level adaptive clipping with adaptive gradient methods (Duchi et al., 2011). These studies provided theoretical utility improvements over DP-SGD under the assumption of gradient magnitude imbalance. While coordinate-level adaptive clipping is conceptually different from iteration-level adaptive clipping, it is feasible to merge these approaches. However, our study primarily focuses on iteration-level adaptive clipping.

Another motivation for adaptive clipping is to reduce the cost of tuning fixed clipping radius (Andrew et al., 2021; Yang et al., 2022; Bu et al., 2022; Xia et al., 2023). Andrew et al. (2021) studied a quantile estimation method to estimate the $L2$ sensitivity of the loss gradient and empirically demonstrated that the hyperparameter of the estimation algorithm was much more robust than using the fixed clipping radius. Yang et al. (2022); Bu et al. (2022); Xia et al. (2023) explored per-sample normalized gradient methods to eliminate the need for a predefined clipping radius and reported that these methods achieved nearly the same accuracy as DP-SGD.

Another related study investigated DP optimization under interpolation regimes. (Asi et al., 2022) proposed a DP optimization based on their so-called Lipschitz extension with a decreasing Lipschitz constant under interpolation regimes for non-federated learning settings. While the theoretical result is quite strong, their theory essentially relies on a quadratic growth condition, which is much stronger than our assumptions, and it requires a huge computational cost due to the need to solve a per-sample optimization sub-problem to obtain the Lipschitz extension at each iteration. In addition, (Béthune et al., 2024) recently proposed a novel algorithm called Clipless DP-SGD, which removes the clipping procedure and its resulting bias in standard DP-SGD by analytically accounting the per-layer Lipschitzness of the neural network during training.

## 2 Notation and Problem Settings

This section first introduces the notation used in this paper. Then, the problem settings are explained.

**Notation.** $\|\cdot\|$ denotes the Euclidean $L_2$ norm $\|\cdot\|_2$: $\|x\| = \sqrt{\sum_i x_i^2}$ for vector $x$. For a natural number $m$, $[m]$ denotes the set $\{1, 2, \ldots, m\}$. For any number $a, b$, $a \vee b$ means $\max\{a, b\}$ and $a \wedge b$ does $\min\{a, b\}$. $\text{Clip}(x, C) := \min\{1, C/\|x\|\}x$ for $x \in \mathbb{R}^d$ and $C \geq 0$. For any $q \in [0, 1]$, $Q_q(X)$ denotes the $q$-quantile of $X$. $\mathbb{R}_{\geq}$ denotes the set of non-negative real numbers.

In this paper, we consider the minimization of objective functions that typically arise in federated learning under differential privacy constraints.

**Objective Function.** The objective function is defined as $f(\cdot) := (1/P) \sum_{p \in [P]} f_p(\cdot)$ on $\mathbb{R}^d$. Here, $f_p(\cdot)$ is the local objective associated with client $p$ and is defined by $(1/n_p) \sum_{z \in D_p} \ell_z(\cdot)$, where $D_p$ is the local dataset of client $p$ and $n_p$ is its size, and $\ell_z$ is the loss function associated with sample $z$. Let $n$ and $n_p$ be $|D|$ and $|D_p|$ respectively.

**Differential Privacy Constraint.** We focus on local record-level differential privacy (also known as Inter-Silo Record-Level (ISRL)-DP (Lowy et al., 2023)), which plays a crucial role in private cross-silo federated learning. Local record-level privacy assumes that both a central server and clients are honest-but-curious, i.e., the server is *not* trustworthy. Then, given a mechanism $\mathcal{M}$, it is required to guarantee record level $(\varepsilon, \delta)$-differential privacy[1] for all the outputs of client $p$ with respect to each local dataset $D_p$ for every $p \in [P]$, that is $\mathbb{P}(\mathcal{M}(D_p) \in S) \leq e^\varepsilon \mathbb{P}(\mathcal{M}(D'_p) \in S) + \delta$, for every $S$ and every adjacent[2]local datasets $D_p$

---

[1]Although we focus on record-level differential privacy, it is straightforward to extend our analysis to the case of client-level differential privacy.

and $D'_p$, where $\varepsilon > 0$ denotes the distinguishable bound of all outputs on two adjacent data subsets and $\delta \in (0, 1)$ represents the probability of information leakage.

We introduce an important notion referred to as the $\tau_q$-quantile-mean ratio, which plays an important role in our theoretical analysis. In the following, let $Q_q$ denote the $q$-quantile of the input values. Note that we always have the loose bound $\tau_q(x) \leq n$.

**Definition 1** ($\tau_q$-quantile-mean ratio). *For given $q \in [0, 1]$, we define $\tau_q : \mathbb{R}^d \to \mathbb{R}_{\geq 0}$ as $\tau_q(x) :=$ $Q_q(\{\|\nabla \ell_z(x)\|^2\}_{z \in D})/((1/n) \sum_{z \in D} \|\nabla \ell_z(x)\|^2)$.*

To analyze the utility of our proposed algorithm, the following theoretical assumptions are required. With the exception of Assumption 2, our assumptions are fairly standard. Importantly, in our DP guarantees analysis, Assumptions 1-4 are not necessary.

**Assumption 1** (Convexity of $\ell_z$). *$\ell_z$ is convex on $\mathbb{R}^d$ for every $z \in D$.*

**Assumption 2** (Interpolation Regimes (Vaswani et al., 2019)). *There exists an optimal solution $x_*$ of $f$, i.e., $x_* \in \arg\min_{x \in \mathbb{R}^d} f(x)$ with $\|x_*\| \leq B$ for some $B > 0$. Moreover, it holds that $\ell_z \geq 0$ and $\ell_z(x_*) = 0$ for every $z \in D$.*

This assumption is a key to our utility improvement. It is well-known that interpolation often occurs in high-dimensional problems including (approximated) kernel methods and over-parameterized deep learning (Jacot et al., 2018; Allen-Zhu et al., 2019). Here, we do not assume the strong growth condition or its variants, which are much stronger than Assumption 2.

**Assumption 3** ($G$-Lipschitzness of $\ell_z$). *$\ell_z$ is $G$-Lipschitz, i.e., $|\ell_z(x) - \ell_z(y)| \leq G\|x - y\|$, for every $x, y \in \mathbb{R}^d$ and $z \in D$.*

**Assumption 4** ($L$-smoothness of $\ell_z$). *$\ell_z$ is $L$-smooth, i.e., $\|\nabla \ell_z(x) - \nabla \ell_z(y)\| \leq L\|x - y\|$, for every $x, y \in \mathbb{R}^d$ and $z \in D$.*

## 3 Approach and Proposed Algorithm

In this section, we present our strategy for incorporating adaptive DP noise in federated learning.

### 3.1 Review of DP-GD and DP-SGD

First, we briefly review the algorithm of DP-GD and DP-SGD (Abadi et al., 2016), which are the gold standard methods in DP optimization based on non-adaptive clipping, to clarify our approach.

In DP optimization, the running algorithm is required to be $(\varepsilon, \delta)$-DP, and a gradient perturbation technique is typically used to satisfy this constraint in federated learning, where privacy preservation is required even during training time communications. Gradient perturbation adds some random noise to the aggregated gradient to satisfy the DP constraint for each iteration. For example, DP-(S)GD has the update rule of $x_r = x_{r-1} - \eta(g_r + \xi_r)$, where $g_r$ is the full gradient $\nabla f(x_{r-1})$ for DP-GD and is a minibatch stochastic gradient for DP-SGD, $\eta$ is a learning rate, and $\xi_r$ is Gaussian noise with mean zero and variance $\sigma^2$, which is called DP noise.

For DP-GD, the noise size $\sigma$ for $(\varepsilon, \delta)$-DP depends on the number of iterations $R$, dataset size $n$, privacy parameters $\varepsilon$, $\delta$, and $L2$ sensitivity $\Delta$ of $g_r$. Specifically, it is known that the DP noise size to satisfy $(\varepsilon, \delta)$-DP is $\sigma = O(\Delta\sqrt{R \log(1/\delta)}/\varepsilon)$. The $L2$ sensitivity of $g_r$ is formally defined as $\Delta := \sup_{d_{\mathrm{H}}(D, D')=1} \|g_r(D) - g_r(D')\|$, where the supremum is taken over every pair of adjacent datasets $(D, D')$ and $g_r(D)$ denotes the gradient computed on dataset $D$.

A standard sensitivity analysis requires the uniform boundedness of the norm of the per-sample gradients, i.e., $\|\nabla \ell_z(x_{r-1})\| \leq C$ for every sample $z \in D$ and every parameter $x_{r-1}$ for some $C \geq 0$. To satisfy this requirement, a typical approach is to use gradient clipping; $g_r := (1/n) \sum_{z \in D} \mathrm{Clip}(\nabla \ell_z(x), C)$ for some predefined $C$. Then, we can get a simple upper bound $\Delta \leq 2C/n$.

---

[2] We say that datasets $D = \{z_i\}_{i=1}^n$ and $D' = \{z'_i\}_{i=1}^n$ are adjacent if $d_{\mathrm{H}}(D, D') = 1$, where $d_{\mathrm{H}}$ is the Hamming distance between $D$ and $D'$ defined by $d_{\mathrm{H}}(D, D') := \sum_{i=1}^n \mathbb{1}_{x_i \neq x'_i}$.

### 3.2 Our Approach

Gradient clipping requires the clipping radius $C$ to execute the algorithm. Most of the previous methods focus on using a constant $C$. However, this may degrade the utility of the algorithm; clipping bias occurs when $\|\nabla\ell_z(x_{r-1})\| \gg C$ for a large potion of $D$, and the privacy budget is wasted in vain when $\|\nabla\ell_z(x_{r-1})\| \ll C$ for every $z \in D$. Thus, it is desirable to adaptively determine $C$ based on the observed per-sample gradients $\{\|\nabla\ell_z(x_{r-1})\|\}_{z\in D}$ for each round $r$ to obtain better utility.

Since $\{\|\nabla\ell_z(x_{r-1})\}_{z\in D}$ depends on $x_{r-1}$ and changes across rounds, it is desirable to make the clipping radius depend on the parameter $x_{r-1}$. This type of clipping is referred to as iteration-level adaptive clipping in this paper. A basic approach to realize iteration-level adaptive clipping is to use the $q$-quantile $Q_q(\{\|\nabla\ell_z(x_{r-1})\|\}_{z\in D})$ with some $q \in [0,1]$ for constructing the clipping radius for each iteration. However, direct use of the quantile is difficult because the quantile information is much more sensitive than the standard aggregation results such as the mean, which results in large DP noise. To mitigate this problem, several estimators have been proposed in the previous study (Fu et al., 2022). However, these are heuristic and do not provide theoretical improvements.

We propose an adaptive clipping radius $C_r$ based on the key quantity $\sqrt{(\tau/n)\sum_{z\in D}\|\nabla\ell_z(x_{r-1})\|^2}$ for a given hyper-parameter $\tau \geq 1$, which can be an upper bound of $Q_q(\{\|\nabla\ell_z(x_{r-1})\|\}_{z\in D)})$ with adequate $\tau$ for some $q \in [0,1]$ with high probability. To guarantee $(\varepsilon,\delta)$-DP of this procedure, we need to add DP noise $\xi_r^{(p),C}$, because $(\tau/n)\sum_{z\in D}\|\nabla\ell_z(x_{r-1})\|^2$ already contains private information. The behind reason for using this quantity is that $(1/n)\sum_{z\in D}\|\nabla\ell_z(x_{r-1})\|^2 \leq 2Lf(x_{r-1})$ holds in general for $L$-smooth convex $f$ in interpolation regimes. This means that $C_r$ can adaptively decrease depending on the current model $x_{r-1}$ as $f(x_{r-1})$ goes to $f(x_*) = 0$ in optimization. Then, the $L2$ sensitivity of $C_r$ and thus the DP noise size also decreases, and we expect better utility than non-adaptive clipping methods.

### 3.3 Proposed Algorithm

---

**Algorithm 1** AdaptDP-FedAvg

---

1: **for** $p \in [P]$ in parallel **do**
2:     Compute $\sigma_p^C$ and $\sigma_p$ based on (1).
3: **end for**
4: $x_0 = x_{\text{ini}}$.
5: **for** $r = 1$ to $R$ **do**
6:     **for** $p \in [P]$ in parallel **do**
7:         $G_r^{(p),C} = \frac{1}{b^C}\sum_{z\in I_r^{(p),C}} \text{Clip}(\|\nabla\ell_z(x_{r-1})\|^2, \hat{G}^2)$, where $I_r^{(p),C} \sim D_p$ with size $b^C$.
8:     **end for**
9:     $C_r = \sqrt{2\tau\left(\frac{1}{P}\sum_{p=1}^P (G_r^{(p),C} + \xi_r^{(p),C}) + \tilde{\nu}\right)} \wedge \hat{G}$, where $\xi_r^{(p),C} \sim N(0, (\sigma_p^C)^2\hat{G}^4)$.
10:     **for** $p \in [P]$ in parallel **do**
11:         $x_{0,r-1}^{(p)} = x_{r-1}$.
12:         **for** $k = 1$ to $K$ **do**
13:             $g_{k,r-1}^{(p)} = \frac{1}{b}\sum_{z\in I_{k,r-1}^{(p)}} \text{Clip}(\nabla\ell_z(x_{k,r-1}^{(p)}), C_r)$, where $I_{k,r-1}^{(p)} \sim D_p$ with size $b$
14:             $x_{k,r-1}^{(p)} = x_{k-1,r-1}^{(p)} - \eta(g_{k,r-1}^{(p)} + \xi_{k,r-1}^{(p)})$, where $\xi_{k,r-1}^{(p)} \sim N(0, \sigma_p^2(C_r)^2 I)$.
15:         **end for**
16:         $\bar{x}_r^{(p)} = \frac{1}{K}\sum_{k=1}^K x_{k-1,r-1}^{(p)}$.
17:     **end for**
18:     $x_r = \text{Clip}(\frac{1}{P}\sum_{p\in[P]} x_{K,r-1}^{(p)}, B)$.
19:     $x_r^{(\text{out})} = \frac{1}{P}\sum_{p\in[P]} \bar{x}_r^{(p)}$.
20: **end for**
21: $x^{(\text{out})} = \frac{1}{R}\sum_{r\in[R]} x_r^{(\text{out})}$.
22: **Return:** $x_R$, $x^{(\text{out})}$.

---

The concrete procedures of our proposed method are given in Algorithm 1. In each round, we first compute $C_r$. Each client $p$ computes $G_r^{(p),C}$ based on local minibatch gradients with size $b^C$, which is the average of the squared per-sample gradient norm with a constant clipping radius $\hat{G}^2$. Then, the central server aggregates $G_r^{(p),C} + \xi_r^{(p),C}$ and constructs $C_r$ as in line 9. Here, $\xi_r^{(p),C} \sim N(0, (\sigma_p^C)^2)$ is Gaussian noise with mean zero and variance $(\sigma_p^C)^2$ is DP noise to protect the privacy of $G_r^{(p),C}$. The constant $\widetilde{\nu}$ is also added to account for the stochastic noise due to the minibatch sampling and the gradient deviation $\|\nabla \ell_z(x_{k-1,r-1}^{(p)}) - \nabla \ell_z(x_{r-1})\|^2$ because we use a common $C_r$ for each client $p$ and each local iteration $k$ within round $r$. Reusing the clipping radius in a round is desirable as it reduces both the number of privacy information releases and the computational cost. After calculating $C_r$, the server sends $C_r$ to each client, and each client runs local optimization based on the adaptive clipping radius $C_r$. The local optimization follows standard DP-SGD with minibatch size $b$ and DP noise $\xi_{k,r-1}^{(p)}$. A key distinction is that $\sigma_r$ depends linearly on the adaptive clipping radius $C_r$ instead of an estimated Lipchitzness parameter $\hat{G}$. The resulting parameters $\{x_{K,r-1}^{(p)}\}_{p \in [P]}$ are aggregated by the server to form the next round global parameter $x_r$. The specific values of $\{\sigma_p^C\}_{p \in [P]}$ and $\{\sigma_p\}_{p \in [P]}$ to guarantee $(\varepsilon, \delta)$-DP will be provided in the next section.

**Remark 1** (Additional computational and communication costs). *Compared to DP-FedAvg, constructing $C_r$ requires additional computational and communication costs. However, these costs are often negligible because (i) $C_r$ is computed only once per round; (ii) $C_r$ allows relying on minibatch sampling, which further reduces the computational cost; and (iii) $G_r^{(p),C}$ is scalar and the communication cost is only $\Theta(P)$ instead of $\Theta(Pd)$, which is quite small.*

**Remark 2** (Hyperparameters). *The hyperparameters of Algorithm 1 are $\eta$, $\tau$ and $\hat{G}$. Since the hyperparameters of DP-SGD are $\eta$ and fixed clipping radius $C$, our algorithm requires only one more hyperparameter than DP-SGD. Note that $\widetilde{\nu}$ can be theoretically determined by Proposition 2 with $u = 0.01$, for example. The parameter clipping radius $B$ never affects the DP guarantees of Algorithm 1 and the clipping procedure is only necessary for technical reasons related to utility analysis. Therefore, in practice, the clipping in line 18 can be ignored.*

## 4 Theoretical Analysis

In this section, we provide a theoretical analysis of our proposed algorithm.

### 4.1 DP Guarantee Analysis

We investigate the DP noise size $\sigma^C$ and $\sigma$ to guarantee $(\varepsilon, \delta)$-DP of Algorithm 1 based on RDP theory (Mironov, 2017) and incorporates the subsampling amplification technique (Wang et al., 2019a).

Given any $\varepsilon : (1, \infty] \to (0, \infty]$ and $\gamma \in (0, 1]$, we define $\mathcal{S}(\varepsilon, \gamma) : (1, \infty] \to (0, \infty]$ as

$$\mathcal{S}(\varepsilon(\cdot), \gamma)(\alpha) := \frac{1}{\alpha - 1} \log \left( 1 + \gamma^2 \binom{\alpha}{2} \min \left\{ 4(e^{\varepsilon(2)} - 1), e^{\varepsilon(2)} \min\{2, (e^{\varepsilon(\infty)} - 1)^2\} \right\} \right.$$
$$\left. + \sum_{j=3}^{\alpha} \gamma^j \binom{\alpha}{j} e^{(j-1)\varepsilon(j)} \min\{2, (e^{\varepsilon(\infty)} - 1)^j\} \right),$$

which is defined in (Wang et al., 2019a). In the following analysis, we assume that the minibatch sampling is based on uniform sampling without replacement.

**Proposition 1** (One-round RDP analysis). *Suppose that $b, b^C \leq n_p$ for $p \in [P]$. Let $\Delta^C := 2/b^C$, $\gamma_p^C := b^C/n_p$, $\Delta := 2/b$ and $\gamma_p := b/n_p$. We define $\varepsilon_p^C(\alpha) := \alpha(\Delta^C)^2/(2(\sigma_p^C)^2)$, $\varepsilon_p(\alpha) := \alpha\Delta^2/(2\sigma_p^2)$. Given $x_{r-1}$, $C_r$ is $(\alpha, \mathcal{S}(\varepsilon_p^C(\cdot), \gamma_p^C)(\alpha))$-RDP, and given $x_{r-1}$ and $C_r$, $(\{x_{k,r-1}^{(p)}\}_{k \in [K]}, \bar{x}_r^{(p)})$ is $(\alpha, K\mathcal{S}(\varepsilon_p(\cdot), \gamma_p)(\alpha))$-RDP for any integer $\alpha \geq 2$.*

**Theorem 1** (DP noise size). *Let $\alpha_* := 1 + \lceil 2\log(1/\delta)/\varepsilon \rceil$. Then, the output information of client $p \in [P]$ in Algorithm 1 is $(\varepsilon, \delta)$-DP, if*

$$
\begin{cases}
\sigma_p^C & := \min\{\sigma_p^C \geq 0 \mid RS(\varepsilon_p^C(\cdot), \gamma_p^C)(\alpha_*) \leq \varepsilon_{\text{target}}\}, \\
\sigma_p & := \min\{\sigma_p \geq 0 \mid RKS(\varepsilon_p(\cdot), \gamma_p)(\alpha_*) \leq \varepsilon_{\text{target}}\}
\end{cases}
\tag{1}
$$

*with $\varepsilon_{\text{target}} := \varepsilon/4$. In particular, the DP noise size $\sigma_p^C$ and $\sigma_p$ in (1) satisfy*

$$
\begin{cases}
\sigma_p^C & = O\left( \dfrac{\sqrt{R\log\frac{1}{\delta}}}{n_p\varepsilon} \vee \dfrac{\sqrt{\log\frac{1}{\delta}}}{b^C\sqrt{\varepsilon}} \right) \\
\sigma_p & = O\left( \dfrac{\sqrt{KR\log\frac{1}{\delta}}}{n_p\varepsilon} \vee \dfrac{\sqrt{\log\frac{1}{\delta}}}{b\sqrt{\varepsilon}} \right)
\end{cases}
\tag{2}
$$

*under $b^C \leq \frac{n_p}{2e\alpha_*} \wedge \left( \frac{4n_p}{\alpha_*(\sigma_p^C)^2} \right)^{\frac{1}{3}}$, $b \leq \frac{n_p}{2e\alpha_*} \wedge \left( \frac{4n_p}{\alpha_*\sigma_p^2} \right)^{\frac{1}{3}}$ and $\varepsilon = O(\log(1/\delta))$.*

**Remark 3.** *When the computation of $C_r$ involves full-batch gradients rather than minibatch ones, replacing $\Delta^C := 2/(b^C)$ and $RS(\varepsilon_p^C(\cdot), \gamma^C)(\alpha_*) \leq \varepsilon_{\text{target}}$ in (1) with $\Delta^C := 2/n_p$ and $R\varepsilon_p^C(\alpha_*) \leq \varepsilon_{\text{target}}$ respectively gives $(\varepsilon, \delta)$-DP guarantees, and (2) still holds.*

### 4.2 Utility Analysis

In this subsection, we derive a convergence rate and utility of Algorithm 1 with $(\varepsilon, \delta)$-DP. In the convergence analysis, the minibatch sampling is assumed to be uniform sampling with replacement for simplicity. Also, to simplify the arguments, we assume that the gradients in the computation of $C_r$ are deterministic, i.e., we focus on the full batch cases, and $n_p = n_{p'}$ holds.

**Notation used in utility analysis.** Let $\ell_z^C(x) := \min\{1, C/\|\nabla\ell_z(x)\|\}\ell_z(x)$, $f_p^C(x) := (1/n_p)\sum_{z \in D_p} \ell_z^C(x)$ and $f^C(x) := (1/P)\sum_{p \in [P]} f_p^C(x)$. Also, $\widehat{\nabla\ell_z^C}(x)$, $\widehat{\nabla f_p^C}(x)$ and $\widehat{\nabla f^C}(x)$ denote $\min\{1, C/\|\nabla\ell_z(x)\|\}\nabla\ell_z(x)$, $(1/n_p)\sum_{z \in D_p} \widehat{\nabla\ell_z^C}(x)$ and $(1/P)\sum_{p \in [P]} \widehat{\nabla f_p^C}(x)$ respectively. We define $\overline{(\sigma^C)^2} := (1/P)\sum_{p=1}^P (\sigma_p^C)^2$, $\overline{\sigma^2} := (1/P)\sum_{p=1}^P \sigma_p^2$, $\bar{\xi}_{r-1}^C := (1/P)\sum_{p=1}^P \xi_{r-1}^{(p),C}$ and $\bar{\xi}_{k,r-1} := (1/P)\sum_{p=1}^P \xi_{k,r-1}^{(p)}$. $n_{\text{loc}}$ denotes $n_p = n_{p'}$.

The following proposition ensures that for at least a $q_\tau(x_{r-1})$-fraction of the $n$ per-sample gradients, our adaptive gradient clipping with radius $C_r$ only reduces to $1/\sqrt{1 \vee (\tau_q(x_{r-1})/\tau)}$ of the original size. Also, it provides a theoretical suggestion for parameter $\widetilde{\nu}$.

**Proposition 2** (Simplified version of Proposition 4). *Let $u \in (0,1)$ be sufficiently small, $q \in [0,1]$ and $\tau > 0$. Suppose that Assumptions 3 and 4 hold. With $\hat{G} \geq G$ and appropriate $\eta \leq \widetilde{O}(1/(KL) \wedge \sqrt{P}/(\sqrt{K\bar{\sigma}^2}dL))$, with probability at least $1 - u$, for every $k \in [K]$, $r \in [R]$, it holds that $Q_q(\{\|\nabla\ell_z(\bar{x}_{k-1,r-1})\|\}_{z \in D}) \leq \sqrt{1 \vee (\tau_q(x_{r-1})/\tau)}C_r$, when we set $\widetilde{\nu} := \sqrt{2\overline{(\sigma^C)^2}\hat{G}^4/P}\sqrt{\log(2PR/u)}$.*

To derive a convergence rate, we first derive a single round bound. Unlike the standard analysis, we develop a bound for $f^{C_r}$ rather than the original $f$. Also, our analysis relies on the arguments from (Woodworth et al., 2020) to bound the localization error due to the use of multiple local updates ($K > 1$).

**Proposition 3** (One-round Analysis (Simplified version of Proposition 5)). *Suppose that Assumptions 1, 2, 3 and 4 hold. Let $u \in (0,1)$ be sufficiently small and $q \in [0,1]$. With probability at least $1 - u$, Algorithm 1 satisfies*

$$
\|\bar{x}_{K,r-1} - x_*\|^2 \leq \|\bar{x}_{0,r-1} - x_*\|^2 - \Omega(\eta)\sum_{k=1}^K \mathbb{E}[f^{C_r}(\bar{x}_{k-1,r-1}) \mid E] + \widetilde{O}(A)C_r^2
$$

$$
-\Omega\left(\frac{\eta}{L}\right)\sum_{k=1}^K \frac{1}{P}\sum_{p \in [P]} \frac{1}{n_p}\sum_{z \in D_p} \frac{1}{\min\{1, C_r/\|\nabla\ell_z(x_{k-1,r-1}^{(p)})\|\}} \|\widehat{\nabla\ell_z^{C_r}}(x_{k-1,r-1}^{(p)})\|^2 + \textit{Stoc. Error.}
$$

*for every $r \in [R]$ under $\tau \leq \tau_1^{\max}$ and $\eta \leq \widetilde{O}(1/(\sqrt{\tau_1^{\max}/\tau}KL) \wedge 1/(\sqrt{\tau_1^{\max}/\tau}\sqrt{K\overline{\sigma^2}d}L))$, where $\tau_q^{\max} :=$ $\max_{r \in [R]} \tau_q(\bar{x}_{r-1})$. Here, $A := \eta^3 K^3 L + \eta^3 K^2 L \overline{\sigma^2} d + (\eta^2 K \overline{\sigma^2} d/P) + (\eta^2 K/(Pb)$ and Stoc. Error. :=$ $-2\eta \sum_{k=1}^{K} \left\langle \bar{x}_{k-1} - x_*, \bar{g}_k - \mathbb{E}_{\{I_k^{(p)}\}_{p \in [P]}}[\bar{g}_k] + \bar{\xi}_{k,r-1} \right\rangle.$*

Then, we incorporate the adaptive clipping radius $C_r$, which includes another DP noise $\xi_r^C$ into consideration in the inequality derived in Proposition 3. Finally, applying this inequality recursively leads to our main result.

**Theorem 2** (Convergence Rate (Simplified version of Theorem 3)). *Suppose that Assumptions 1, 2, 3 and 4 hold. Let $u \in (0,1)$ be sufficiently small. Then, with probability at least $1 - u$, Algorithm 1 with $\widetilde{\nu}$ as defined in Proposition 2 and $(\sigma^C, \sigma)$ as defined in Theorem 4, satisfies*

$$\frac{1}{KR} \sum_{r=1}^{R} \sum_{k=1}^{K} f^{C_r}(\bar{x}_{k-1,r-1})$$
$$\leq \widetilde{O}\left( \frac{\|x_{\text{ini}} - x_*\|^2 + B^2}{\eta KR} + \frac{\tau\sqrt{R}G^2}{n_{\text{loc}}\sqrt{P}\varepsilon} \left( \eta^2 K^2 L + \frac{\eta KRd}{n_{\text{loc}}^2 P\varepsilon^2} + \frac{\eta}{Pb} + \left( \eta KL + \frac{1}{P} \right) \frac{\eta d}{b^2\varepsilon} \right) \right).$$

*under $\hat{G} \geq G$ with $\hat{G} = \Theta(G)$, $\tau \leq \tau_1^{\max}$ and*

$$\eta \leq \widetilde{O}\left( \frac{1}{\phi(\tau)KL} \wedge \frac{1}{KGB} \wedge \frac{n_{\text{loc}}\varepsilon}{\phi(\tau)K\sqrt{Rd}L} \wedge \frac{n_{\text{loc}}^2 P\varepsilon^2}{\psi(\tau)KRLd} \wedge \frac{Pb}{\psi(\tau)L} \right),$$

*where $\phi(\tau) := \sqrt{\tau_1^{\max}/\tau} + (\tau_1^{\max}\tau)^{1/4}$, $\psi(\tau) := \sqrt{\tau_1^{\max}\tau}$ and $\tau_q^{\max}$ is defined in Proposition 3.*

Note that Theorem 2 only guarantees the convergence of $f^{C_r}$ rather than $f$. To obtain meaningful results, we use a fact that there exists $D^q \subset D$ such that $|D^q| \geq qn$ and $f_{D^q}(\bar{x}_{k-1,r-1}) \leq O(\sqrt{1 \vee (\tau_q^{\max})/\tau})f^{C_r}(\bar{x}_{k-1,r-1})$ under suitable conditions, where $f_{D^q}(x) := (1/|D^q|)\sum_{z \in D^q} \ell_z(x)$. Note that when $q = 1$, $D^1 = D$ and thus we can always have a lower bound of $f^{C_r}$ based on $f$. Using this fact, from Theorem 2, choosing appropriate $\eta$, $K$ and $R$, which minimize the convergence error yields the best achievable utility of Algorithm 1 with $(\varepsilon, \delta)$-DP guarantees, summarized in the following corollary.

**Corollary 1** (Utility bound (Simplified version of Corollary 2)). *It is assumed that $\|x_{\text{ini}} - x_*\|$, $B$, $G$ and $L$ are $\Theta(1)$. Then, for any $q \in [0.5, 1.0]$, there exists $D_q \subset D$ such that $|D_q| \geq q|D|$ and Algorithm 1 with $(\varepsilon, \delta)$-DP guarantees satisfies*

$$f_{D^q}\left( x^{(\text{out})} \right)$$
$$\leq \widetilde{O}\left( \sqrt{1 \vee \frac{\tau_q^{\max}}{\tau}} \left( \frac{\tau^{\frac{4}{9}}}{d^{\frac{2}{9}}}(\epsilon_{\text{util}}^{DP\text{-}SGD})^{\frac{10}{9}} + \frac{(\phi(\tau)\tau\sqrt{P})^{\frac{1}{3}}}{d^{\frac{1}{6}}}(\epsilon_{\text{util}}^{DP\text{-}SGD})^{\frac{4}{3}} + \frac{\sqrt{\tau}}{d^{\frac{1}{4}}}(\epsilon_{\text{util}}^{DP\text{-}SGD})^{\frac{3}{2}} + \frac{\tau^{\frac{7}{15}}}{d^{\frac{7}{30}}}(\epsilon_{\text{util}}^{DP\text{-}SGD})^{\frac{19}{15}} \right. \right.$$
$$\left. \left. + \frac{(\phi(\tau)\tau^2\sqrt{P})^{\frac{1}{5}}}{d^{\frac{1}{5}}}(\epsilon_{\text{util}}^{DP\text{-}SGD})^{\frac{7}{5}} + \frac{\psi(\tau)^{\frac{1}{5}}}{d^{\frac{1}{5}}}(\epsilon_{\text{util}}^{DP\text{-}SGD})^{\frac{8}{5}} + \frac{\tau^{\frac{2}{5}}}{(\phi(\tau)d)^{\frac{1}{5}}}(\epsilon_{\text{util}}^{DP\text{-}SGD})^{\frac{6}{5}} + \psi(\tau)(\epsilon_{\text{util}}^{DP\text{-}SGD})^2 \right) \right).$$

*under the conditions in Theorem 2, where $f_{D^q}(x) := (1/|D^q|)\sum_{z \in D^q} \ell_z(x)$, $\epsilon_{\text{util}}^{DP\text{-}SGD} = \widetilde{\Theta}(\sqrt{d}/(n_{\text{loc}}\sqrt{P}\varepsilon))$ and $\phi(\tau), \psi(\tau)$ are defined in Theorem 2.*

**Remark 4** (Interpretation of Corollary 1). *Corollary 1 gives a utility bound of $f_{D^q}$, which represents the averaged risk on some dataset $D^q \subset D$ with $|D^q| \geq qn$. At first, setting $q = 1$ gives a utility bound of the original objective $f$, since $D^1 = D$ always holds. In this case, setting $\tau := \Theta(\tau_1^{\max})$ minimizes the utility bound. However, setting $q < 1$ can yield a much better bound because $\tau_q^{\max}$ may be much smaller than $\tau_1^{\max}$, at the expense of the replacement of $f$ with $f_{D^q}$. This observation suggests that using $\tau \ll \tau_1^{\max}$ can be beneficial in practical situations. This point will be further explored in Section 5. Note that our algorithm does not explicitly depend on the choice of $q$, the bound given in Corollary 1 holds every $q \in [0.5, 1.0]$ simultaneously.*

**Remark 5** (Comparison with the best known utility bound). *The best known utility under local record-level $(\varepsilon, \delta)$-DP is $\epsilon_{\text{util}}^{DP\text{-}SGD} := \widetilde{\Theta}(\sqrt{d}/(n_{\text{loc}}\sqrt{P}\varepsilon))$ for non-strongly convex objectives, which is achieved by DP-(S)GD. Compared with this rate, we can see that all the terms of our bound have $o(\epsilon_{\text{util}}^{DP\text{-}SGD})$, thanks to the interpolation property and our adaptive clipping scheme. Thus, our method can achieve better theoretical utility than the best-known one of DP-(S)GD in our problem settings. Although each term has a factor depending on $\tau$, $\tau_q^{\max}$ and $\tau_1^{\max}$, it also has a factor of $1/d^a$ for some $a > 0$ except for the last term, which is favorable in high-dimensional problems. In Section 5, we will empirically confirm that $\tau_q^{\max}$ is not significantly large.*

## 5 Numerical Experiments

### 5.1 Experimental Setups

To empirically demonstrate the efficiency of the proposed AdaptDP-FedAvg, as detailed in Algorithm 1, several experiments were conducted.

**Comparing Algorithms** For comparison, we use DP-FedAvg (Abadi et al., 2016) with a fixed noise size as the baseline. However, the original analysis of DP-FedAvg did not incorporate the subsampling amplification technique, which we include in our analysis in Section 4. To ensure a fair comparison, we conduct a new study to determine the appropriate noise size for DP-FedAvg, along with its update rules, as presented in Algorithm 2 in Appendix D.

**Task Settings** To compare the performance of algorithms—AdaptDP-FedAvg using adaptive noise size versus DP-FedAvg employing fixed noise size—we prepared three privacy levels for each dataset, as shown in Table 1. Based on our understanding, conventional studies on model training using DP typically follow one of two evaluation approaches: i) setting the target privacy level $(\varepsilon, \delta)$ first and then computing the corresponding noise size $\sigma$, e.g., (Pichapati et al., 2019; Geyer et al., 2017; Triastcyn & Faltings, 2019); or ii) setting the noise size $\sigma$ in advance, which implicitly determines the resulting $(\varepsilon, \delta)$, e.g., (Abadi et al., 2016; Bu et al., 2020; Fu et al., 2022). While the number of data samples and rounds needed for model convergence vary across datasets, the latter approach is more practical for testing purposes, and we therefore adopted it. As shown in Table 1, we first set three fixed noise sizes for DP-FedAvg on each dataset. For a fair comparison, the corresponding privacy level $\varepsilon$ was computed while fixing $\delta = 10^{-4}$, using (3) in Appendix D. Then, $(\sigma_p, \sigma_p^C)$ for AdaptDP-FedAvg was determined, as in line 2 in Algorithm 1. Note that noise sizes differ between AdaptDP-FedAvg and DP-FedAvg as in Table 1, the same privacy level $(\varepsilon, \delta)$ is theoretically ensured.

In total, we prepared four benchmark tests, denoted T1)–T4). For T1), which uses an artificial dataset, we evaluated the function $1/(2n_{\text{train}})(x - b)^{\top}(Z^{\top}Z)(x - b)$, where $b$ represents the reference parameters, $x$ the model parameters, and $Z$ the data samples. The dataset was randomly generated with $Z \in \mathbb{R}^{d \times n_{\text{train}}}$. In addition, we utilized real-world datasets, including T2) Bank Marketing[3] ($n_{\text{train}} = 45,211$), T3) MNIST ($n_{\text{train}} = 3,000$), and T4) FashionMNIST ($n_{\text{train}} = 3,000$) (Xiao et al., 2017). Separate test datasets were prepared for each task.

For the convex model, we employed a two-layer Multi-Layer Perceptron (MLP) with fixed (i.e., untrainable) first-layer weights to ensure convexity. This design choice is motivated by our utility analysis in Corollary 1, which suggests that a larger model parameter size $d$ can lead to a smaller upper bound on the utility for convex objectives. In our setup, the hidden layer dimension was set to 128 units for T2) and 512 units for T3) - T4). The training dataset, consisting of $n_{\text{train}}$ samples, was partitioned across $P = 2$ clients such that each client holds $n_{\min} = \lfloor n_{\text{train}}/P \rfloor$ samples. Test accuracy was recorded at the center server at the end of each round. As additional experiments, we conducted evaluations with $P = 4, 6$ clients, as well as experiments using a non-convex model. These results are provided in Appendix E.2 and Appendix E.3, respectively.

**Hyperparameter Tuning** We configured $(R, K)$ for each dataset to ensure that the model parameters approached convergence using a minibatch size of $b = 100$ and $\tau = 1$. Under this condition, we performed

---

[3]https://archive.ics.uci.edu/dataset/222/bank+marketing

Table 1: Comparison of noise sizes for each privacy level and tested hyperparameters used for tuning, across each method and dataset. Noise sizes in AdaptDP-FedAvg and DP-FedAvg are chosen to ensure the same privacy level $(\varepsilon, \delta)$, as described in Section 5.1.

| Dataset | Privacy level | DP-FedAvg | AdaptDP-FedAvg | Tested hyperparameters | |
|---|---|---|---|---|---|
| | | | | $\eta$ | $(C, \hat{G})$ |
| T1) Artificial dataset | Lv.1 | $\sigma_p = 0.003$ | $(\sigma_p, \sigma_p^C) = (0.0042, 0.00097)$ | | |
| | Lv.2 | $\sigma_p = 0.006$ | $(\sigma_p, \sigma_p^C) = (0.0078, 0.0021)$ | $\{0.1, 0.3, 0.5\}$ | $\{0.5, 1, 3, 5\}$ |
| | Lv.3 | $\sigma_p = 0.01$ | $(\sigma_p, \sigma_p^C) = (0.012, 0.0043)$ | | |
| T2) Bank Marketing | Lv.1 | $\sigma_p = 0.006$ | $(\sigma_p, \sigma_p^C) = (0.0077, 0.0025)$ | | |
| | Lv.2 | $\sigma_p = 0.01$ | $(\sigma_p, \sigma_p^C) = (0.0103, 0.0078)$ | $\{0.01, 0.03, 0.1\}$ | $\{0.05, 0.1, 0.5\}$ |
| | Lv.3 | $\sigma_p = 0.03$ | $(\sigma_p, \sigma_p^C) = (0.0524, 0.010)$ | | |
| T3) MNIST | Lv.1 | $\sigma_p = 0.003$ | $(\sigma_p, \sigma_p^C) = (0.0041, 0.0011)$ | | |
| | Lv.2 | $\sigma_p = 0.006$ | $(\sigma_p, \sigma_p^C) = (0.093, 0.0021)$ | $\{0.1, 0.3\}$ | $\{0.05, 0.1, 0.5\}$ |
| | Lv.3 | $\sigma_p = 0.01$ | $(\sigma_p, \sigma_p^C) = (0.013, 0.0050)$ | | |
| T4) FashionMNIST | Lv.1 | $\sigma_p = 0.003$ | $(\sigma_p, \sigma_p^C) = (0.0041, 0.0011)$ | | |
| | Lv.2 | $\sigma_p = 0.006$ | $(\sigma_p, \sigma_p^C) = (0.093, 0.0021)$ | $\{0.03, 0.1, 0.3\}$ | $\{0.05, 0.1, 0.5\}$ |
| | Lv.3 | $\sigma_p = 0.01$ | $(\sigma_p, \sigma_p^C) = (0.013, 0.0050)$ | | |

Table 2: The lowest training loss and the highest test accuracy over $R$ rounds with selected hyperparameters. For T1), test accuracy is omitted since it is not a classification task.

| Dataset | Privacy level | DP-FedAvg Trainloss/TestAcc | Selected $(C, \eta)$ | AdaptDP-FedAvg Trainloss/TestAcc | Selected $(\hat{G}, \eta)$ |
|---|---|---|---|---|---|
| T1) Artificial dataset | Lv.1 | $6.73e^{-6}$ ±0.28 /− | $(0.5, 0.3)$ | $\mathbf{3.34e^{-7}}$ ±0.43 /− | $(1, 0.1)$ |
| $(R, K){=}(150, 20)$ | Lv.2 | $2.69e^{-5}$ ±0.21 /− | $(0.5, 0.3)$ | $\mathbf{3.81e^{-6}}$ ±0.15 /− | $(3, 0.1)$ |
| | Lv.3 | $7.62e^{-5}$ ±0.24 /− | $(0.5, 0.3)$ | $\mathbf{1.82e^{-5}}$ ±0.10 /− | $(3, 0.1)$ |
| T2) Bank Marketing | Lv.1 | $3.13e^{-1}$±0.02/89.08±0.21 | $(0.05, 0.03)$ | $\mathbf{3.12e^{-1}}$ ±0.02 /$\mathbf{89.43}$ ±0.14 | $(0.05, 0.03)$ |
| $(R, K){=}(100, 226)$ | Lv.2 | $3.12e^{-1}$±0.17/$\mathbf{89.26}$±0.33 | $(0.5, 0.03)$ | $3.12e^{-1}$±0.04/88.73 ±0.42 | $(0.1, 0.03)$ |
| | Lv.3 | $3.13e^{-1}$±0.15/89.09±0.40 | $(0.05, 0.03)$ | $3.13e^{-1}$±0.13/$\mathbf{89.15}$±0.16 | $(0.1, 0.03)$ |
| T3) MNIST | Lv.1 | $4.84e^{-5}$±0.10 /$\mathbf{61.50}$ ±5.50 | $(0.05, 0.3)$ | $\mathbf{4.25e^{-5}}$ ±0.41/56.50 ±0.50 | $(0.05, 0.3)$ |
| $(R, K) = (1000, 15)$ | Lv.2 | $5.57e^{-5}$ ±0.31/$\mathbf{64.75}$ ±7.25 | $(0.05, 0.3)$ | $\mathbf{4.38e^{-5}}$ ±0.53/59.00 ±1.00 | $(0.05, 0.3)$ |
| | Lv.3 | $4.06e^{-4}$ ±1.44/57.35 ±0.35 | $(0.1, 0.1)$ | $\mathbf{4.02e^{-5}}$ ±0.09/$\mathbf{60.30}$ ±1.20 | $(0.05, 0.3)$ |
| T4) FashionMNIST | Lv.1 | $2.02e^{-2}$ ±1.97/57.50 ±7.50 | $(0.05, 0.3)$ | $\mathbf{5.96e^{-5}}$ ±4.51/$\mathbf{68.50}$ ±7.50 | $(0.1, 0.3)$ |
| $(R, K){=}(1000, 15)$ | Lv.2 | $3.39e^{-2}$ ±1.10/60.00 ±4.00 | $(0.5, 0.03)$ | $\mathbf{2.94e^{-4}}$ ±2.30/$\mathbf{67.20}$ ±7.80 | $(0.1, 0.3)$ |
| | Lv.3 | $1.28e^{-1}$ ±0.22/29.50 ±2.50 | $(0.1, 0.03)$ | $\mathbf{1.27e^{-1}}$ ±0.05/$\mathbf{32.80}$ ±0.70 | $(0.05, 0.3)$ |

experiments with several hyperparameter settings, as shown in Table 1. The results obtained with the optimal combination of $(C, \hat{G})$ and $\eta$ to achieve the lowest training loss are summarized in Subsec. 5.2. Further details are provided in Appendix E.1.

## 5.2 Experimental Results

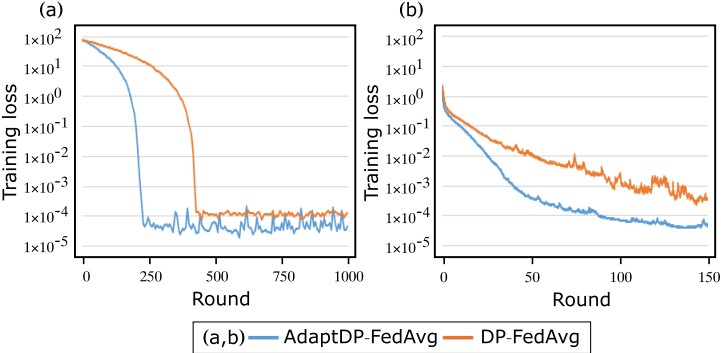

Figure 1: Training loss curves with selected/tuned hyperparameters. (a) corresponds to the T1) artificial dataset, while (b) corresponds to T3) MNIST under privacy level Lv.3.

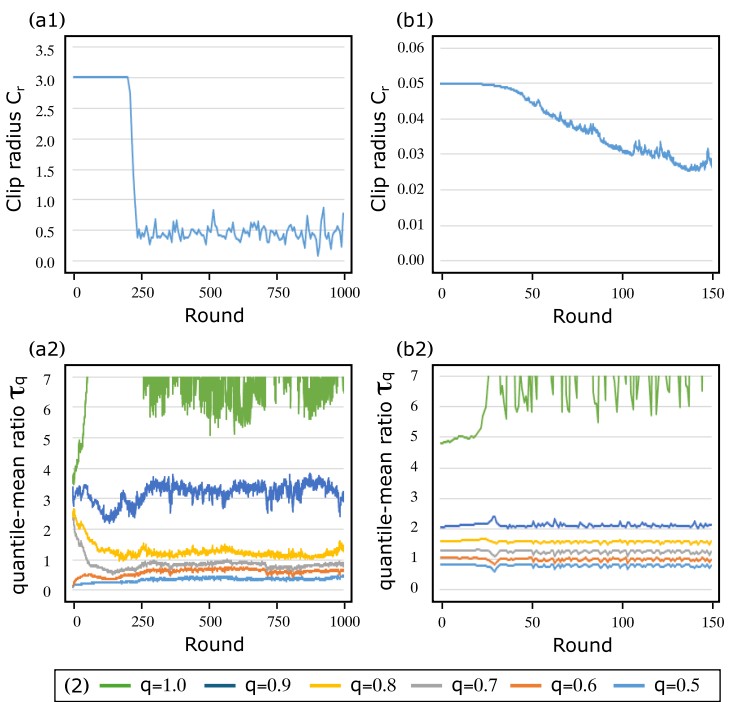

Figure 2: (a1) Transition of clipping radius $C_r$ and (a2) transition of $\tau_q$ for various $q = \{0.5(\text{median}), \dots, 1.0\ (\text{max})\}$ in AdaptDP-FedAvg with well-tuned hyperparamers on T1) artificial dataset under privacy level Lv.3. (b1) and (b2) present the corresponding results for T3) MNIST under privacy level Lv.3.

Table 2 presents the experimental results obtained using the selected hyperparameter settings that achieve the lowest training loss for each method. A comprehensive version of the experimental results, including various hyperparameter settings (e.g., learning rate $\eta$, noise size $\sigma$, clip radius $C, \hat{G}$), is summarized in Table 3 in Appendix E.1. From Table 2, we observe that AdaptDP-FedAvg consistently achieved lower training loss than DP-FedAvg. Lower training loss generally corresponds to smaller gradient norms, which we numerically confirm in the following paragraph for a part of tasks. Since the noise size in AdaptDP-FedAvg is determined by the gradient norm, it typically injects less noise size compared to DP-FedAvg. This reduced noise injection may be a key factor contributing to the superior performance of AdaptDP-FedAvg. Thus, AdaptDP-FedAvg was empirically effective under the conditions of lower training loss (i.e., smaller gradient norms). In some cases, the test accuracy of AdaptDP-FedAvg is lower than that of DP-FedAvg, which is likely due to overfitting. Note that our theory primarily focuses on effectively reducing training loss, and low test accuracy in some experimental settings does not contradict our theory. It should also be noted that the variance of DP-FedAvg is very large (e.g., 5.5 and 7.25), whereas the variance of AdaptDP-FedAvg remains small. While DP-FedAvg may appear to achieve better average performance, this is likely due to a few seeds that happened to yield exceptionally good results. Therefore, it does not indicate that DP-FedAvg consistently outperforms AdaptDP-FedAvg.

To further analyze the experimental results, Fig. 1 presents the convergence curves using training loss. Fig. 2 shows (a1) transitions of the adaptive clipping radius $C_r$ in AdaptDP-FedAvg and (a2) transitions of the quantile-mean ratio $\tau_q$ for T1) artificial dataset, while the corresponding results for T3) MNIST are shown in Figs. 2 (b1) and (b2), respectively. As shown in Figs. 2 (a2) and (b2), $\tau_q$ is approximately 1.0 when $q = 0.7$, indicating that 70% of data samples have gradient norms smaller than the average gradient norm. Furthermore, when $q \geq 0.8$, $\tau_q$ does not diverge but instead converges to a moderate value. This observation supports Corollary 1, which theoretically asserts that the training loss can be lower compared to DP-SGD. Empirically, as shown in Fig. 1, the training loss of AdaptDP-FedAvg is lower than that of DP-FedAvg. This improvement can be attributed to the adaptive changes in the clipping radius $C_r$ as the communication round progresses, as illustrated in Figs. 2(a1) and (b1). In contrast, DP-FedAvg employs a fixed clipping radius

throughout the training process. When a large proportion of data samples undergoes gradient clipping, performance will be degraded. However, this result suggests that as the training progresses, the number of data samples that remain unclipped increases, leading to improved performance of AdaptDP-FedAvg. Moreover, through our experiments, we obtained empirical insights indicating that $\tau = 1$ and approximately $\hat{G} = 0.05$ tend to yield good performance for the models and datasets we used. However, the optimal values of $\tau$ and $\hat{G}$ vary depending on the models and datasets, and there is currently no principled method to determine them universally. Similar to learning rate selection, a greedy search remains necessary.

## 6 Conclusion

We proposed AdaptDP-FedAvg, a method designed to enhance the utility of standard differential private optimization algorithms by introducing adaptive clipping radius in federated learning. Our adaptive clipping radius was determined based on the root-mean-square of the gradient norms, motivated by the interpolation property and smoothness of the objectives. We theoretically demonstrated the privacy analysis of the proposed algorithm in general settings. Furthermore, for smooth and non-strongly convex objectives, we showed that our method achieves an improved utility bound compared to DP-SGD in specific cases. Through several numerical experiments, we confirmed that our AdaptDP-FedAvg outperforms standard DP optimization (DP-FedAvg) with a fixed clipping radius.

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

# A Preliminary

## A.1 Review of Rényi Differential Privacy

Our DP analysis relies on Rényi differential privacy (RDP) technique. This sub-section is excerpted from (Murata & Suzuki, 2023), which summarizes several known results about RDP used in our theoretical analysis.

**Definition 2** $((\alpha, \varepsilon)$-RDP (Definition 4 in Mironov (2017))). *A randomized mechanism $\mathcal{M} : \mathcal{D} \to \mathcal{R}$ satisfies $(\alpha, \varepsilon)$-RDP ($\alpha \in (1, \infty)$ and $\varepsilon > 0$) if for any datasets $D, D' \in \mathcal{D}$ with $d_{\mathrm{H}}(D, D') = 1$, it holds that*

$$\frac{1}{\alpha - 1} \log \mathbb{E}_{o \sim \mathcal{M}(D')} \left( \frac{\mathcal{M}(D)(o)}{\mathcal{M}(D')(o)} \right)^{\alpha} \leq \varepsilon,$$

*where $M(D)(o)$ denotes the density of $\mathcal{M}(D)$ at $o$.*

**Lemma 1** (Post-processing Property of RDP (Mironov (2017))). *Let $\mathcal{M} : \mathcal{D} \to \mathcal{R}$ be $(\alpha, \varepsilon)$-RDP and $g : \mathcal{R} \to \mathcal{R}'$ be any function. Then, $g \circ \mathcal{M} : \mathcal{D} \to \mathcal{R}'$ is also $(\alpha, \varepsilon)$-RDP.*

**Lemma 2** (Composition of RDP Mechanisms (Proposition 1 in Mironov (2017))). *Let $\mathcal{M}_r : \mathcal{R}_1 \times \cdots \times \mathcal{R}_{r-1} \times \mathcal{D} \to \mathcal{R}_r$ be $(\alpha, \varepsilon_r)$-RDP for $r \in [R]$. Then, $\mathcal{M} : \mathcal{D} \to \mathcal{R}_1 \times \cdots \times \mathcal{R}_R$ defined by $\mathcal{M}(D) := (\mathcal{M}_1(D), \mathcal{M}_2(\mathcal{M}_1(D), D), \ldots, \mathcal{M}_R(\mathcal{M}_1(D), \ldots, D))$ is $(\alpha, \sum_{r=1}^{R} \varepsilon_r)$-RDP.*

**Lemma 3** (From RDP to DP (Proposition 3 in Mironov (2017))). *If a randomized mechanism $\mathcal{M}$ is $(\alpha, \varepsilon)$-RDP, then $\mathcal{M}$ is $(\varepsilon + \log(1/\delta)/(\alpha - 1), \delta)$-DP for every $\delta \in (0, 1)$.*

**Definition 3** ($l_q$-sensitivity). *$\Delta_q(h) := \sup_{d_{\mathrm{H}}(D, D') = 1} \|h(D) - h(D')\|_q$ is called $l_q$-sensitivity for function $h$, where the maximum is taken over any adjacent datasets $D, D' \in \mathcal{D}$.*

**Lemma 4** (Gaussian Mechanism (Corollary 3 in Mironov (2017))). *Given a function $h$, Gaussian Mechanism $\mathcal{M}(D) := h(D) + \mathcal{N}(0, \sigma_1^2 I)$ satisfies $(\alpha, \alpha \Delta_2^2(h)/(2\sigma_1^2))$-RDP for every $\alpha \in (1, \infty)$.*

**Lemma 5** (Subsampling Amplification (Theorem 9 in (Wang et al., 2019a))). *Let $\mathcal{M}$ be a randomized mechanism that takes a dataset of $b \leq n$ points as an input and $\gamma := b/n$. $\mathcal{M} \circ \mathrm{subsample}_\gamma$ be defined as: (1) $\mathrm{subsample}_\gamma$: subsample $\gamma n$ points without replacement from the input dataset with size $n$, and (2) apply $\mathcal{M}$ taking the subsampled points as the input. For every integer $\alpha \geq 2$, if $\mathcal{M}$ is $(\alpha, \varepsilon(\alpha))$-RDP, $\mathcal{M} \circ \mathrm{subsample}_\gamma$ is $(\alpha, \varepsilon'(\alpha))$-RDP, where*

$$\varepsilon'(\alpha) \leq \frac{1}{\alpha - 1} \log \left( 1 + \gamma^2 \binom{\alpha}{2} \min \left\{ 4(e^{\varepsilon(2)} - 1), e^{\varepsilon(2)} \min\{2, (e^{\varepsilon(\infty)} - 1)^2\} \right\} \right.$$
$$\left. + \sum_{j=3}^{\alpha} \gamma^j \binom{\alpha}{j} e^{(j-1)\varepsilon(j)} \min\{2, (e^{\varepsilon(\infty)} - 1)^j\} \right).$$

We can derive a simple upper bound using Lemma 5.

**Lemma 6** (Subsampling Amplification (Simple Upper Bound)). *On the same settings as in Lemma 5, if $\varepsilon(\alpha)$ is monotonically increasing with respect to $\alpha$, it holds that*

$$\varepsilon'(\alpha) \leq \frac{2}{3} \left( 4 + \frac{e}{c} \right) \frac{\gamma^2 \alpha^2 \varepsilon(2)}{\alpha - 1}$$

*under $\varepsilon(\alpha) \leq 1/3 \wedge \log(1/(2\gamma\alpha))$ and $\gamma \leq \varepsilon(2)/(c\alpha)$ for any $c > 0$.*

*Proof.* See the proof of Lemma B.8 of (Murata & Suzuki, 2023). $\square$

## A.2 Concentration Inequalities

In this subsection, we provide known concentration inequalities for norm-subGaussian random vectors (Jin et al., 2019) used in our analysis. Note that any random vector $X$ satisfying $\|X\| \leq \sigma$ is $2\sigma$-norm-subGaussian (and any $\sigma$-bounded centered random vector is $\sigma$-norm-subGaussian). Thus, the following concentration inequalities is also applicable to bounded random vectors.

**Lemma 7** (Martingle version of Azume-Hoeffding's inequality for norm-subGaussian (Corollary 7 in Jin et al. (2019))). *Let $X_1, \ldots, X_n$ be random vectors in $\mathbb{R}^d$. Suppose that $\{X_i\}_{i=1}^n$ and corresponding filtrations $\{\mathfrak{F}_i\}_{i=1}^n$ satisfy the following conditions:*

$$\mathbb{E}[X_i \mid \mathfrak{F}_{i-1}] = 0 \ and \ \mathbb{P}(\|X_i\| \geq s \mid \mathfrak{F}_{i-1}) \leq 2e^{-\frac{s^2}{2\sigma_i^2}}, \forall s \in \mathbb{R}, \forall i \in [n]$$

*for fixed $\{\sigma_i\}_{i=1}^n$. Then, for any $u \in (0,1)$, with probability at least $1-u$ it holds that*

$$\left\| \sum_{i=1}^n X_i \right\| \leq c\sqrt{\sum_{i=1}^n \sigma_i^2 \log \frac{2d}{u}},$$

*where $c := \sqrt{2}$.*

**Lemma 8** (Martingle version of Azume-Hoeffding's inequality for norm-subGaussian (Jin et al., 2019)). *Let $X_1, \ldots, X_n$ be random vectors in $\mathbb{R}^d$. Suppose that $\{X_i\}_{i=1}^n$ and corresponding filtrations $\{\mathfrak{F}_i\}_{i=1}^n$ satisfy the following conditions:*

$$\mathbb{E}[X_i \mid \mathfrak{F}_{i-1}] = 0 \ and \ \mathbb{P}(\|X_i\| \geq s \mid \mathfrak{F}_{i-1}) \leq 2e^{-\frac{s^2}{2\sigma_i^2}}, \forall s \in \mathbb{R}, \forall i \in [n]$$

*for random variables $\{\sigma_i\}_{i=1}^n$ with $\sigma_i \in \mathfrak{F}_{i-1}(i \in [n])$. Then, for any $u \in (0,1)$ and $A > a > 0$, with probability at least $1-q$ it holds that*

$$\sum_{i=1}^n \sigma_i^2 \geq A \ or \ \left\| \sum_{i=1}^n X_i \right\| \leq c\sqrt{\max\left\{ \sum_{i=1}^n \sigma_i^2, a \right\} \left( \log \frac{2d}{u} + \log\log \frac{A}{a} \right)},$$

*where $c := \sqrt{2}$.*

**Remark 6.** *In Lemma 8, in practice, we can choose sufficiently small $a > 0$ and a deterministic value $A > a$ such that $\sum_{i=1}^n \sigma_i^2 < A$ and then we have essentially the same bound as in Lemma 7, because $1/a$ and $A$ will always have only polynomial dependencies on the other parameters such as $K$, $R$, $P$, $n$, $G$, $L$, $C_{\text{proj}}$, $d$, $1/\varepsilon$ and $1/\delta$ in our analysis. This approach sacrifices some rigor but yields correct results and simplifies the analysis.*

# B   DP Guarantee Analysis

## B.1   Proof of Proposition 6

Here, we focus on DP analysis with respect to client $p \in [P]$.

### RDP Analysis of $C_r$

Let minibatch $I_r^{(p),C}$ be given. Then, we can see that the $L2$ sensitivity with respect to client $p \in [P]$ of $G_r^{(p)}$ is $\Delta^C \hat{G}^2$, where $\Delta^C := \frac{2}{b^C}$, because $|G_r^{(p),C}(I) - G_r^{(p),C}(I')| \leq 2\hat{G}^2/b^C$ for adjacent minibatches $I$ and $I'$, where $G_r^{(p),C}(I)$ denotes the averaged squared per-sample gradient norm on the samples $I$. From Lemmas 4 and 1, we conclude that $C_r$ is $(\alpha, \varepsilon_p^C(\alpha))$-RDP, where $\varepsilon_p^C(\alpha) := \alpha(\Delta^C)^2/(2(\sigma_p^C)^2)$.

Then, we apply the subsampling amplification technique to the proven results for the fixed minibatch. From Lemma 5, it holds that for every integer $\alpha \geq 2$, $C_r$ is $(\alpha, \mathcal{S}(\varepsilon_p^C(\cdot), \gamma_p^C)(\alpha))$-RDP with $\gamma_p^C := b^C/n_p$.

### RDP Analysis of $\{x_{k,r-1}^{(p)}\}_{k \in [K]}$

In the following, we omit round index $r$. We consider RDP property of $x_k^{(p)}$ conditioned on $C_r$ and $x_{k-1}^{(p)}$. To realize this, first, we focus on the $L2$ sensitivity of $g_k^{(p)}$ conditioned the previous randomness. Since $g_k^{(p)}$ is

constituted of $b$ samples, we have that $\|g_k^{(p)}(I) - g_k^{(p)}(I')\| \leq 2C_r/b$ for adjacent minibatches $I$ and $I'$, where $g_k^{(p)}(I)$ denotes the averaged gradient on the samples $I$. Thus, the mechanism that takes a minibatch with size $b$ as an input and returns the sum of the average of the clipped per-sample gradients and Gaussian noise $\xi_k^{(p)} \sim N(0, \sigma_p^2 C_r^2 I)$ leads to $(\alpha, \varepsilon_p(\alpha))$-RDP for every $\alpha \in (1, \infty)$ from Lemma 4, where $\varepsilon_p(\alpha) := 2\alpha/(b^2\sigma_p^2)$. Then, from Lemma 5 and Lemma 1, taking into account the subsampling amplification effect, we immediately obtain that $x_k^{(p)}$ is $(\alpha, \mathcal{S}(\varepsilon_p(\cdot), \gamma_p)(\alpha))$-RDP for every integer $\alpha \geq 2$ with $\gamma_p := b/n_p$.

Using the composition theorem of RDP (Lemma 2) finishes the proof of Proposition 6. $\qquad\square$

### B.2   Proof of Theorem 4

Let $\alpha \geq 2$ be some integer. Combining Proposition 6 with Lemma 2, it holds that the outputs of client $p \in [P]$ is $(\alpha, RS(\varepsilon_p^C(\cdot), \gamma_p^C)(\alpha) + RKS(\varepsilon_p(\cdot), \gamma_p)(\alpha))$-RDP. From Lemma 3, we can see that the outputs of client $p \in [P]$ is $(\varepsilon/2 + \log(1/\delta)/(\alpha - 1), \delta)$-DP under $RS(\varepsilon_p^C(\cdot), \gamma_p^C)(\alpha)$, $RKS(\varepsilon_p(\cdot), \gamma_p)(\alpha) \leq \varepsilon/4 = \varepsilon_{\text{target}}$. Thus, using $\alpha_* := 1 + \lceil 2\log(1/\delta)/\varepsilon \rceil$ implies $(\varepsilon, \delta)$-DP. Thus, we only need to determine the minimum noise size $\sigma_p^C$ and $\sigma_p$ satisfying $RS(\varepsilon_p^C(\cdot), \gamma_p^C)(\alpha)(\alpha_*) \leq \varepsilon_{\text{target}}$ and $RKS(\varepsilon_p(\cdot), \gamma_p)(\alpha_*) \leq \varepsilon_{\text{target}}$ as in (1).

Finally, we estimate the order of the noise size. Then, we only need to determine the minimum noise size $\sigma_p$ satisfying $KR\varepsilon_{\gamma_p}(\alpha_*) \leq \varepsilon_{\text{target}}$ and $\sigma_p^C$ satisfying $R\varepsilon_{\gamma_p^C}^C(\alpha_*) \leq \varepsilon_{\text{target}}$ as in (1). We first estimate the order of the noise size $\sigma_p$. Suppose that

$$\sigma_p^2 = \Theta\left(\frac{KR\log\frac{1}{\delta}}{(n_p\varepsilon)^2} \vee \frac{\log\frac{1}{\delta}}{b^2\varepsilon}\right)$$

is sufficiently large and

$$b \leq \frac{n_p}{2e\alpha_*} \wedge \left(\frac{4n_p}{\alpha_*\sigma_p^2}\right)^{\frac{1}{3}}.$$

Since $2\gamma_p\alpha_* \leq 1/e$, it holds that $\log(1/(2\gamma_p\alpha_*)) \geq 1$. Thus, $\varepsilon(\alpha_*) = 2\alpha_*/(b^2\sigma_p^2) \leq 1/3 \wedge \log(1/(2\gamma_p\alpha_*))$ is satisfied under $\varepsilon = O(\log(1/\delta))$. Also, note that $\gamma_p \leq 4/(\alpha_*b^2\sigma_p^2) = \varepsilon(2)/\alpha_*$ holds. Thus, the conditions of Lemma 6 are satisfied. Finally, from Lemma 6, we have

$$\varepsilon'_{\gamma_p}(\alpha_*) \leq O\left(\frac{\gamma_p^2\alpha_*^2\varepsilon_p(2)}{\alpha_* - 1}\right) = O\left(\frac{\alpha_*}{n_p^2\sigma_p^2}\right)$$

under $\varepsilon = O(\log(1/\delta))$.

Thus, choosing appropriate $\sigma_p$ gives $KRS(\varepsilon_p^C(\cdot), \gamma_p^C)(\alpha_*) \leq \varepsilon_{\text{target}}$.

Similarly, setting

$$(\sigma_p^C)^2 = \Theta\left(\frac{R\log\frac{1}{\delta}}{(n_p\varepsilon)^2} \vee \frac{\log\frac{1}{\delta}}{(b^C)^2\varepsilon}\right)$$

is sufficiently large and

$$b^C \leq \frac{n_p}{2e\alpha_*} \wedge \left(\frac{4n_p}{\alpha_*(\sigma_p^C)^2}\right)^{\frac{1}{3}}$$

gives $RS(\varepsilon_p^C(\cdot), \gamma_p^C)(\alpha_*) \leq \varepsilon_{\text{target}}$

This finishes the proof of Theorem 4. $\qquad\square$

## C   Utility Analysis

In this section, we provided utility analysis of Algorithm 1. All the minibatch sampling is assumed to be uniform sampling with replacement. Also, to simplify the arguments, we assume that the gradients in the

computation of $C_r$ are deterministic, i.e., we focus on the full batch cases. Moreover, we assume that $n_p = n_{p'}$ for every $p, p' \in [P]$ and define $n_{\text{loc}} := n_p$.

First, we give some notation used in our analysis.

**Notation**

Let $\ell_z^C(x) := \min\{1, C/\|\nabla \ell_z(x)\|\} \ell_z(x)$, $f_p^C(x) := (1/n_p) \sum_{z \in D_p} \ell_z^C(x)$ and $f^C(x) := (1/P) \sum_{p \in [P]} f_p^C(x)$. Also, $\widehat{\nabla \ell_z^C}(x)$ denotes $\min\{1, C/\|\nabla \ell_z(x)\|\} \nabla \ell_z(x)$. Similarly, $\widehat{\nabla f_p^C}(x)$ denotes $(1/n_p) \sum_{z \in D_p} \widehat{\nabla \ell_z^C}(x)$ and $\widehat{\nabla f^C}(x)$ denotes $(1/P) \sum_{p \in [P]} \widehat{\nabla f_p^C}(x)$. Note hat $\widehat{\nabla \ell_z^C}(x) \neq \nabla \ell_z^C(x)$ in general.

Also, we define $\bar{\xi}_r^C := (1/P) \sum_{p=1}^P \xi_r^{(p),C}$, $\bar{\xi}_{k,r-1} := (1/P) \sum_{p=1}^P \xi_{k,r-1}^{(p),C}$, $\overline{(\sigma^C)^2} := (1/P) \sum_{p=1}^P (\sigma_p^C)^2$ and $\overline{\sigma^2} := (1/P) \sum_{p=1}^P \sigma_p^2$.

As long as no confusion, we omit the round index $r$. For example, we write $x_{k,r-1}^{(p)}$ as $x_k^{(p)}$ in $r$-th round of Algorithm 1.

## C.1 Useful Concentration Results

**Lemma 9.** *Suppose that Assumption 3 holds. Let $u \in (0, 1)$. With probability at least $1 - u$, for every $k \in [K]$, $r \in [R]$, and $p \in [P]$, it holds that*

$$\left| \sum_{\kappa=1}^k \frac{1}{b} \sum_{z \in I_\kappa^{(p)}} \langle \nabla f(x_{\kappa-1}^{(p)}), \text{Clip}(\nabla \ell_z(x_{\kappa-1}^{(p)}), C_r) - \frac{1}{n_p} \sum_{z \in D_p} \text{Clip}(\nabla \ell_z(x_{\kappa-1}^{(p)}), C_r) \rangle \right|$$
$$\leq \widetilde{O}\left( \frac{\sqrt{K} G C_r}{\sqrt{b}} \right).$$

*Proof.* Observe that $|\langle \nabla f(x_{\kappa-1}^{(p)}), \text{Clip}(\nabla \ell_z(x_{\kappa-1}^{(p)}), C_r) - \frac{1}{n_p} \sum_{z \in D_p} \text{Clip}(\nabla \ell_z(x_{\kappa-1}^{(p)}), C_r) \rangle| \leq 2GC_r$. Applying Lemma 8 with $A = 2KbG$ and sufficiently small $a > 0$ and taking union bounds over $k \in [K]$, $p \in [P]$ and $r \in [R]$ give the desired result. $\square$

**Lemma 10.** *Let $u \in (0, 1)$. With probability at least $1 - u$, for every $k \in [K]$, $r \in [R]$, and $p \in [P]$, it holds that*

$$\left\| \frac{1}{b} \sum_{z \in I_k^{(p)}} \text{Clip}(\nabla \ell_z(x_{k-1}^{(p)}), C_r) - \frac{1}{n_p} \sum_{z \in D_p} \text{Clip}(\nabla \ell_z(x_{k-1}^{(p)}), C_r) \right\| \leq \widetilde{O}\left( \frac{C_r}{\sqrt{b}} \right).$$

*Similarly, with probability at least $1 - u$, for every $k \in [K]$, $r \in [R]$, it holds that*

$$\left\| \frac{1}{P} \sum_{p \in [P]} \frac{1}{b} \sum_{z \in I_k^{(p)}} \text{Clip}(\nabla \ell_z(x_{k-1}^{(p)}), C_r) - \frac{1}{n_p} \sum_{z \in D_p} \text{Clip}(\nabla \ell_z(x_{k-1}^{(p)}), C_r) \right\| \leq \widetilde{O}\left( \frac{C_r}{\sqrt{Pb}} \right).$$

*Proof.* Applying Lemma 7 and taking union bounds over $p \in [P]$, $k \in [K]$ and $r \in [R]$ give the desired results. $\square$

**Lemma 11.** *Let $u \in (0, 1/4)$. With probability at least $1 - 4u$, for every $p \in [P]$, $k \in [K]$ and $r \in [R]$, it holds that*

$$|\xi_r^{(p),C}| \leq \widetilde{O}(\sigma_p^C \hat{G}^2), \quad |\bar{\xi}_r^C| \leq \widetilde{O}\left( \sqrt{\frac{\overline{(\sigma^C)^2}}{P}} \hat{G}^2 \right),$$

$$\|\xi_{k,r-1}^{(p)}\| \leq \widetilde{O}(\sigma_p C_r \sqrt{d}), \quad \|\bar{\xi}_{k,r-1}\| \leq \widetilde{O}\left(\sqrt{\frac{\sigma^2 d}{P}} C_r\right)$$

*and*

$$\mathrm{Clip}(g_{k,r-1}^{(p)} + \xi_{k,r-1}^{(p)}, C_{\mathrm{proj}}) = g_{k,r-1}^{(p)} + \xi_{k,r-1}^{(p)}$$

*under* $C_{\mathrm{proj}} \geq \widetilde{\Omega}((1 + \sigma_p \sqrt{d})\hat{G})$.

*Proof.* Since $\xi_r^{(p),C} \sim N(0, \sigma_p^C \hat{G}^2)$, we have $\mathbb{P}(|\xi_r^{(p),C}| > s) \leq 2\exp(-s^2/(2(\sigma_p^C \hat{G}^2)^2))$ for any $s > 0$. This implies $|\xi_r^C| \leq \sqrt{2}\sigma^C \hat{G}^2 \sqrt{\log(2/u)}$ with probability at least $1 - u$. Taking union bounds over $p \in [P]$ and $r \in [R]$ gives the desired result. The proof of the case of $\bar{\xi}_r^C$ is perfectly same. Thus, the first result is proven. Similarly, we have $(\xi_{k,r-1}^{(p)})_j^2 \leq 2\sigma_p^2 C_r^2 \log(2/u)$ with probability at least $1 - u$. Taking union bounds with respect to $j \in [d]$, $p \in [P]$, $k \in [K]$ and $r \in [R]$ gives the desired result. The case of $\bar{\xi}_{k,r-1}$ is completely same. This gives the second result. The last result is trivial since $C_r \leq \hat{G}$. $\square$

In the following, we always assume that the events defined in Lemmas 9, 10 and 11, which simultaneously hold with probability at least $1 - 7u$ for any $u \in (0, 1/7)$.

**Remark**

Due to a technical reason, we modify Algorithm 1 of line 16 as follows: $x_{k,r-1}^{(p)} = x_{k-1,r-1}^{(p)} - \eta\mathrm{Clip}(g_{k,r-1}^{(p)} + \xi_{k,r-1}^{(p)}, C_{\mathrm{proj}})$ for $C_{\mathrm{proj}} > 0$. First note that this modification does not affect the DP guarantee analysis thanks to the post processing property of RDP. We particularly use sufficiently large $C_{\mathrm{proj}} := \widetilde{\Theta}((1 + \max_{p \in [P]} \sigma_p \sqrt{d})\hat{G})$. Under this choice, we can see that (i) the sequence of model parameters $\{x_{k,r-1}\}_{k \in [K], r \in [R]}$ is bounded almost surely, which is necessary to utlize Lemma 8, and (ii) the clipping does not happen for every $k$ and $r$ with high probability from Lemma 11. Practically, we do not need to implement this clipping process thanks to (ii). Additionally, the choice of $C_{\mathrm{proj}}$ is quite robust because $C_{\mathrm{proj}}$ only depends on the convergence rate with log-log order. Thus, we can omit this clipping process in practice as in Algorithm 1.

## C.2 Analysis of Clipping Radius

**Lemma 12.** *Suppose that Assumption 4 holds. For any $x_1, x_2 \in \mathbb{R}^d$ and $C > 0$, we have*

$$\min\{1, C/\|\nabla\ell_z(x_1)\|\} \leq \min\{1, C/\|\nabla\ell_z(x_2)\|\} + \frac{L\|x_1 - x_2\|}{C}.$$

*Proof.* We will prove the lemma by considering different cases below

**(i)** $\|\nabla\ell_z(x_2)\| \leq C$

In this case, $\min\{1, C/\|\nabla\ell_z(x_1)\|\} \leq 1 = \min\{1, C/\|\nabla\ell_z(x_2)\|\}$ holds.

**(ii)** $\|\nabla\ell_z(x_2)\| > C$ **and** $\|\nabla\ell_z(x_1)\| \leq C$

Observe that

$$
\begin{aligned}
|\min\{1, C/\|\nabla\ell_z(x_2)\|\} - 1| &= \frac{\|\nabla\ell_z(x_2)\| - C}{\|\nabla\ell_z(x_2)\|} \\
&\leq \frac{\|\nabla\ell_z(x_2)\| - \|\nabla\ell_z(x_1)\|}{C} \\
&\leq \frac{L\|x_1 - x_2\|}{C}
\end{aligned}
$$

and thus $\min\{1, C/\|\nabla\ell_z(x_1)\|\} \leq \min\{1, C/\|\nabla\ell_z(x_2)\|\} + \frac{L\|x_1-x_2\|}{C}$.

**(iii)** $\|\nabla\ell_z(x_2)\| > C$ **and** $\|\nabla\ell_z(x_1)\| > C$

In this case, we have

$$
\begin{aligned}
|\min\{1, C/\|\nabla\ell_z(x_1)\|\} - \min\{1, C/\|\nabla\ell_z(x_2)\|\}| &= C\frac{|\|\nabla\ell_z(x_1)\| - \|\nabla\ell_z(x_2)\|\|}{\|\nabla\ell_z(x_1)\|\|\nabla\ell_z(x_2)\|} \\
&\leq \frac{|\|\nabla\ell_z(x_1)\| - \|\nabla\ell_z(x_2)\|\|}{C}.
\end{aligned}
$$

The remained proof is similar to the arguments of case (ii). $\qquad\square$

**Lemma 13.** *Let $u \in (0, 1/11)$. Suppose that the requirements in Proposition 2 are satisfied. Then, with probability $1 - 11u$, for every $k \in [K]$, $r \in [R]$ and $x \in \mathbb{R}^d$, it holds that*

$$
\|\bar{x}_k - \bar{x}_0\|^2 \leq \widetilde{O}\left(\frac{\eta^2 K C_r^2}{Pb} + \frac{\eta^2 K \overline{\sigma^2} d C_r^2}{P}\right) + O(\eta^2 K)\sum_{k=1}^{K}\left\|\frac{1}{P}\sum_{p\in[P]}\widehat{\nabla f_p^{C_r}}(x_{k-1}^{(p)})\right\|^2.
$$

*In particular, we have*

$$
\|\bar{x}_k - \bar{x}_0\|^2 \leq \widetilde{O}\left(\eta^2 K^2 C_r^2 + \frac{\eta^2 K \overline{\sigma^2} d C_r^2}{P}\right).
$$

*Here, $\bar{x}_k := \frac{1}{P}\sum_{p\in[P]} x_k^{(p)}$.*

*Similarly, we have*

$$
\|x_k^{(p)} - x_0^{(p)}\|^2 \leq \widetilde{O}\left(\eta^2 K^2 C_r^2 + \eta^2 K \sigma_p^2 d C_r^2\right)
$$

*for every $p \in [P]$.*

*Proof.* Let $r$ be fixed and we omit index $r$. We define $\bar{g}_k := \frac{1}{P}\sum_{p\in[P]} g_k^{(p)}$ and $\bar{\xi}_k := \frac{1}{P}\sum_{p\in[P]} \xi_k^{(p)}$. Observe that

$$
\begin{aligned}
\|\bar{x}_k - \bar{x}_0\|^2 &= \|\bar{x}_{k-1} - \bar{x}_0\|^2 + \eta\langle\bar{x}_{k-1} - \bar{x}_0, \bar{x}_k - \bar{x}_{k-1}\rangle + \eta^2\|\bar{x}_k - \bar{x}_{k-1}\|^2 \\
&\leq \left(1 + \frac{1}{K}\right)\|\bar{x}_{k-1} - \bar{x}_0\|^2 - 2\eta\langle\bar{x}_{k-1} - \bar{x}_0, \mathbb{E}[\bar{g}_k]\rangle \\
&\quad - \frac{1}{K}\|\bar{x}_{k-1} - \bar{x}_0\|^2 - 2\eta\langle\bar{x}_{k-1} - \bar{x}_0, \bar{g}_k - \mathbb{E}[\bar{g}_k]\rangle - 2\eta\langle\bar{x}_{k-1} - \bar{x}_0, \bar{\xi}_k\rangle \\
&\quad + O(\eta^2)\|\mathbb{E}[g_k]\|^2 + O(\eta^2)\|g_k - \mathbb{E}[g_k]\|^2 \\
&\leq \left(1 + \frac{1}{K}\right)\|\bar{x}_{k-1} - \bar{x}_0\|^2 + O(\eta^2 K)\|\mathbb{E}[\bar{g}_k]\|^2 \\
&\quad - \frac{1}{2K}\|\bar{x}_{k-1} - \bar{x}_0\|^2 - 2\eta\langle\bar{x}_{k-1} - \bar{x}_0, \bar{g}_k - \mathbb{E}[\bar{g}_k]\rangle - 2\eta\langle\bar{x}_{k-1} - \bar{x}_0, \bar{\xi}_k\rangle \\
&\quad + O(\eta^2)\|g_k - \mathbb{E}[g_k]\|^2 \\
&\leq \left(1 + \frac{1}{K}\right)\|\bar{x}_{k-1} - \bar{x}_0\|^2 \\
&\quad - \frac{1}{2K}\|\bar{x}_{k-1} - \bar{x}_0\|^2 - 2\eta\langle\bar{x}_{k-1} - \bar{x}_0, \bar{g}_k - \mathbb{E}[\bar{g}_k]\rangle - 2\eta\langle\bar{x}_{k-1} - \bar{x}_0, \bar{\xi}_k\rangle \\
&\quad + O(\eta^2 K)\left\|\frac{1}{P}\sum_{p\in[P]}\widehat{\nabla f_p^{C_r}}(x_{k-1}^{(p)})\right\|^2 + \widetilde{O}\left(\frac{\eta^2 C_r^2}{Pb}\right).
\end{aligned}
$$

Here, for the last inequality, we used Lemma 10.

Now, we use this inequality recursively. Since $(1 + 1/K)^k \le e = O(1)$, we have

$$\|\bar{x}_k - \bar{x}_0\|^2$$
$$\le -\frac{1}{2K} \sum_{\kappa=1}^{k} \|\bar{x}_{\kappa-1} - \bar{x}_0\|^2 - 2\eta \sum_{\kappa=1}^{k} v(\kappa) \langle \bar{x}_{\kappa-1} - \bar{x}_0, \bar{g}_k - \mathbb{E}[\bar{g}_k] \rangle - 2\eta \sum_{\kappa=1}^{k} v(\kappa) \langle \bar{x}_{\kappa-1} - \bar{x}_0, \bar{\xi}_k \rangle$$
$$+ O(\eta^2 K) \sum_{k=1}^{K} \left\| \frac{1}{P} \sum_{p \in [P]} \widehat{\nabla f_p^{C_r^{(p)}}}(x_{k-1}^{(p)}) \right\|^2 + \widetilde{O}\left( \frac{\eta^2 K C_r^2}{Pb} \right),$$

where $v(\kappa) := (1 + 1/K)^{k-\kappa}$.

Then, applying Lemma 8 to the third and fourth terms of the right hand side, with probability $1 - 2u$, we get

$$\left| 2\eta \sum_{\kappa=1}^{k} v(\kappa) \langle \bar{x}_{\kappa-1} - \bar{x}_0, \bar{g}_k - \mathbb{E}[\bar{g}_k] \rangle \right| \le \widetilde{O}\left( 2\eta \sqrt{\sum_{\kappa=1}^{k} v(\kappa)^2 \|\bar{x}_{\kappa-1} - \bar{x}_0\|^2 \frac{C_r^2}{Pb}} \right)$$
$$\le \widetilde{O}\left( \frac{\eta^2 K C_r^2}{Pb} \right) + \frac{1}{4K} \sum_{\kappa=1}^{k} \|\bar{x}_{\kappa-1} - x_0\|^2$$

and

$$\left| 2\eta \sum_{\kappa=1}^{k} v(\kappa) \langle \bar{x}_{\kappa-1} - \bar{x}_0, \bar{\xi}_k \rangle \right| \le \widetilde{O}\left( 2\eta \sqrt{\sum_{\kappa=1}^{k} v(\kappa)^2 \|\bar{x}_{\kappa-1} - \bar{x}_0\|^2 \frac{\overline{\sigma^2} C_r^2 d}{P}} \right)$$
$$\le \widetilde{O}\left( \frac{\eta^2 K \overline{\sigma^2} C_r^2 d}{P} \right) + \frac{1}{4K} \sum_{\kappa=1}^{k} \|\bar{x}_{\kappa-1} - x_0\|^2,$$

where $\overline{\sigma^2} := (1/P) \sum_{p=1}^{P} \sigma_p^2$. Here, note that it holds that $\|\bar{x}_{\kappa-1} - \bar{x}_0\| \le \eta K C_{\text{proj}}$.

Hence, we obtain

$$\|\bar{x}_k - \bar{x}_0\|^2 \le \widetilde{O}\left( \frac{\eta^2 K C_r^2}{Pb} + \frac{\eta^2 K \overline{\sigma^2} d C_r^2}{P} \right) + O(\eta^2 K) \sum_{k=1}^{K} \left\| \frac{1}{P} \sum_{p \in [P]} \widehat{\nabla f_p^{C_r^{(p)}}}(x_{k-1}^{(p)}) \right\|^2.$$

Noting $\|\widehat{\nabla f_p^{C_r}}(x_{k-1}^{(p)})\| \le C_r$ finishes the proof of the first result. The proof of the second result is completely same as that of the first result. □

**Proposition 4.** *Let $u \in (0, 1/11)$, $q \in [0,1]$ and $\tau > 0$. Suppose that Assumptions 3 and 4 hold. Under $\hat{G} \ge G$ and $\eta \le O(1/(KL) \wedge \sqrt{P}/(\sqrt{K\overline{\sigma^2}dL}))$, with probability at least $1 - 11u$, for every $k \in [K]$, $r \in [R]$, it holds that*

$$Q_q(\{\|\nabla \ell_z(\bar{x}_{k-1,r-1})\|\}_{z \in D}) \le \sqrt{1 \vee \frac{\tau_q(x_{r-1})}{\tau}} C_r,$$

*when we define*

$$C_r := \sqrt{2\tau \left( \frac{1}{n} \sum_{z \in D} \text{Clip}(\nabla \|\ell_z(x_{r-1})\|^2, \hat{G}^2) + \widetilde{\nu} + \bar{\xi}_r \right)} \wedge \hat{G},$$

*where*

$$\widetilde{\nu} := c\sqrt{\frac{\overline{(\sigma^C)^2}}{P}\hat{G}^2}\sqrt{\log\frac{2PR}{u}}$$

*and $\bar{\xi}_r := (1/P)\sum_{p=1}^P \xi_r^{(p)}$. Here, $c = \sqrt{2}$.*

*Proof.* Let $p$, $k$ and $r$ be fixed. $x_{k-1}^{(p)}$ denotes $x_{k,r-1}^{(p)}$. From Assumption 4, we have $\|\nabla\ell_z(\bar{x}_{k-1})\|^2 \le (1+\gamma)\|\nabla\ell_z(\bar{x}_0)\|^2 + \left(1+\frac{1}{\gamma}\right)\|\nabla\ell_z(\bar{x}_{k-1}) - \nabla\ell_z(\bar{x}_0)\|^2 \le (1+\gamma)\|\nabla\ell_z(\bar{x}_0)\|^2 + L^2\left(1+\frac{1}{\gamma}\right)\|\bar{x}_{k-1} - \bar{x}_0\|^2$ for any $z \in D$, $p \in [P]$ and $\gamma > 0$. Also, note that $\text{Clip}(\nabla\ell_z(x_{k-1}^{(p)}), \hat{G}) = \nabla\ell_z(x_{k-1}^{(p)})$ from Assumption 3 and $\hat{G} \ge G$. Thus, we have

$$\begin{aligned}
&Q_q(\{\|\nabla\ell_z(\bar{x}_{k-1})\|^2\}_{z\in D})\\
&\le \frac{(1+\gamma)\tau_q(\bar{x}_0)}{P}\sum_{p\in[P]}\frac{1}{n_p}\sum_{z\in D_p}\|\nabla\ell_z(x_{r-1})\|^2 + \left(1+\frac{1}{\gamma}\right)L^2\|\bar{x}_{k-1} - \bar{x}_0\|^2\\
&\le \frac{(1+\gamma)\tau_q(\bar{x}_0)}{P}\sum_{p\in[P]}\frac{1}{n_p}\sum_{z\in D_p}\|\nabla\ell_z(x_{r-1})\|^2 + \left(1+\frac{1}{\gamma}\right)\tilde{O}\left(\eta^2 K^2 L^2 + \frac{\eta^2 KL^2\bar{\sigma}^2 d}{P}\right)C_r^2\\
&\le \frac{(1+\gamma)\tau_q(\bar{x}_0)}{P}\sum_{p\in[P]}\frac{1}{n_p}\sum_{z\in D_p}\|\nabla\ell_z(x_{r-1})\|^2 + \frac{\left(1+\frac{1}{\gamma}\right)C_r^2}{16}
\end{aligned}$$

for some $p \in [P]$. Here, the third inequality holds from Lemma 13. The last inequality holds by choosing appropriate $\eta \le O(1/(KL) \wedge \sqrt{P}/(\sqrt{K\bar{\sigma}^2 d}L))$.

Thus, setting $\gamma := 1/3$ gives

$$\begin{aligned}
Q_q(\{\|\nabla\ell_z(\bar{x}_{k-1})\|^2\}_{z\in D}) &\le \frac{1+\gamma}{2}\frac{\tau_q(x_{r-1})}{\tau}C_r^2 + \frac{\left(1+\frac{1}{\gamma}\right)C_r^2}{16}\\
&\le \max\left\{1, \frac{\tau_q(x_{r-1})}{\tau}\right\}C_r^2.
\end{aligned}$$

Here, we used the fact that $\widetilde{\nu} \ge \bar{\xi}_r$ from 11.

This finishes the proof of Proposition 2. $\qquad\square$

## C.3 Convergence Analysis

**Proposition 5** (One-round Analysis)**.** *Suppose that Assumptions 1, 2, 3 and 4 hold. Let $u \in (0, 1/11)$ and $q \in [0, 1]$. With probability at least $1 - 11u$, Algorithm 1 satisfies*

$$\|\bar{x}_{K,r-1} - x_*\|^2$$

$$\leq \|\bar{x}_{0,r-1} - x_*\|^2 - \Omega(\eta) \sum_{k=1}^{K} \mathbb{E}[f^{C_r}(\bar{x}_{k-1,r-1}) \mid E]$$

$$+ \widetilde{O}\left(\eta^3 K^3 L + \eta^3 K^2 L \overline{\sigma^2} d + \frac{\eta^2 K \overline{\sigma^2} d}{P} + \frac{\eta^2 K}{Pb}\right) C_r^2$$

$$- \Omega\left(\frac{\eta}{L}\right) \sum_{k=1}^{K} \frac{1}{P} \sum_{p \in [P]} \frac{1}{n_p} \sum_{z \in D_p} \frac{1}{\min\{1, C_r/\|\nabla \ell_z(x_{k-1,r-1}^{(p)})\|\}} \|\widehat{\nabla \ell_z^{C_r}}(x_{k-1,r-1}^{(p)})\|^2$$

$$- 2\eta \sum_{k=1}^{K} \left\langle \bar{x}_{k-1,r-1} - x_*, \bar{g}_{k,r-1} - \mathbb{E}_{\{I_{k,r-1}^{(p)}\}_{p \in [P]}}[\bar{g}_{k,r-1}] \right\rangle$$

$$- 2\eta \sum_{k=1}^{K} \langle \bar{x}_{k-1,r-1} - x_*, \bar{\xi}_{k,r-1} \rangle.$$

*for every* $r \in [R]$ *under* $\tau \leq \tau_1^{\max}$ *and* $\eta \leq \widetilde{O}(1/(\sqrt{\tau_1^{\max}/\tau}KL) \wedge 1/(\sqrt{\tau_1^{\max}/\tau}\sqrt{K\sigma_p^2 d}L))$, *where* $\tau_q^{\max} := \max_{r \in [R]} \tau_q(\bar{x}_{r-1})$.

*Proof.* First note that from Lemma 11, it holds that $\text{Clip}(g_{k,r-1}^{(p)} + \xi_{k,r-1}^{(p)}, C_{\text{proj}}) = g_{k,r-1}^{(p)} + \xi_{k,r-1}^{(p)}$ with high probability.

We define $\bar{x}_k := \frac{1}{P} \sum_{p \in [P]} x_k^{(p)}$, $\bar{g}_k := \frac{1}{P} \sum_{p \in [P]} g_k^{(p)}$. Since $\mathbb{E}_{\{I_k^{(p)}\}_{p \in [P]}}[\bar{g}_k] = (1/P) \sum_{p \in [P]} \widehat{\nabla f_p^{C_r}}(x_{k-1}^{(p)})$, we have

$$\|\bar{x}_k - x_*\|^2$$

$$= \|\bar{x}_{k-1} - x_*\|^2 - 2\eta\langle \bar{x}_{k-1} - x_*, \bar{g}_k \rangle - 2\eta\langle \bar{x}_{k-1} - x_*, \bar{\xi}_k \rangle + 2\eta^2 \|\bar{g}_k\|^2 + 2\eta^2 \|\bar{\xi}_k\|^2$$

$$\leq \|\bar{x}_{k-1} - x_*\|^2 - 2\eta \left\langle \bar{x}_{k-1} - x_*, \frac{1}{P} \sum_{p \in [P]} \widehat{\nabla f_p^{C_r}}(x_{k-1}^{(p)}) \right\rangle + 4\eta^2 \left\| \frac{1}{P} \sum_{p \in [P]} \widehat{\nabla f_p^{C_r}}(x_{k-1}^{(p)}) \right\|^2$$

$$- 2\eta \left\langle \bar{x}_{k-1} - x_*, \bar{g}_k - \mathbb{E}_{\{I_k^{(p)}\}_{p \in [P]}}[\bar{g}_k] \right\rangle - 2\eta\langle \bar{x}_{k-1} - x_*, \bar{\xi}_k \rangle$$

$$+ 4\eta^2 \left\| \bar{g}_k - \mathbb{E}_{\{I_k^{(p)}\}_{p \in [P]}}[\bar{g}_k] \right\|^2 + 2\eta^2 \|\bar{\xi}_k\|^2.$$

Note that

$$- \left\langle \bar{x}_{k-1} - x_*, \frac{1}{P} \sum_{p \in [P]} \widehat{\nabla f_p^{C_r}}(x_{k-1}^{(p)}) \right\rangle$$

$$= \frac{1}{P} \sum_{p \in [P]} \frac{1}{n_p} \sum_{z \in D_p} \left( \left\langle x_{k-1}^{(p)} - \bar{x}_{k-1}, \widehat{\nabla \ell_z^{C_r}}(x_{k-1}^{(p)}) \right\rangle + \left\langle x_* - x_{k-1}^{(p)}, \widehat{\nabla \ell_z^{C_r}}(x_{k-1}^{(p)}) \right\rangle \right).$$

Since

$$\left\langle x_{k-1}^{(p)} - \bar{x}_{k-1}, \nabla \ell_z(x_{k-1}^{(p)}) \right\rangle \leq \ell_z(x_{k-1}^{(p)}) - \ell_z(\bar{x}_{k-1}) + \frac{L}{2} \|x_{k-1}^{(p)} - \bar{x}_{k-1}\|^2$$

from Assumption 4 and

$$\left\langle x_* - x_{k-1}^{(p)}, \nabla \ell_z(x_{k-1}^{(p)}) \right\rangle \leq \ell_z(x_*) - \ell_z(x_{k-1}^{(p)}) - \frac{1}{2L} \|\nabla \ell_z(x_{k-1}^{(p)})\|^2$$

from Assumptions 1, 2 and 4, we get

$$-\left\langle \bar{x}_{k-1} - x_*, \nabla \ell_z(x_{k-1}^{(p)}) \right\rangle \leq -\ell_z(\bar{x}_{k-1}) + \frac{L}{2}\|x_{k-1}^{(p)} - \bar{x}_{k-1}\|^2 - \frac{1}{2L}\|\nabla \ell_z(x_{k-1}^{(p)})\|^2.$$

Multiplying the both side by $\min\{1, C_r/\|\nabla \ell_z(x_{k-1}^{(p)})\|\} \leq 1$, we have

$$-\left\langle \bar{x}_{k-1} - x_*, \widehat{\nabla \ell_z^{C_r}}(x_{k-1}^{(p)}) \right\rangle$$
$$\leq -\min\{1, C_r/\|\nabla \ell_z(x_{k-1}^{(p)})\|\}\ell_z(\bar{x}_{k-1}) + \frac{L}{2}\|x_{k-1}^{(p)} - \bar{x}_{k-1}\|^2 - \frac{1}{2\min\{1, C_r/\|\nabla \ell_z(x_{k-1}^{(p)})\|\}L}\|\widehat{\nabla \ell_z^{C_r}}(x_{k-1}^{(p)})\|^2.$$

Now we bound the first term. Suppose that From Proposition 2, for any $z \in D_p$ and $p \in [P]$, we have $\|\nabla \ell_z(\bar{x}_{k-1})\| \leq \sqrt{1 \vee (\tau_1(\bar{x}_{r-1})/\tau)}C_r$, and thus it holds that

$$\min\{1, C_r/\|\nabla \ell_z(\bar{x}_{k-1})\|\} \geq 1/\sqrt{1 \vee (\tau_1(\bar{x}_{r-1})/\tau)}.$$

Choosing appropriate $\eta \leq \widetilde{O}(1/(\sqrt{1 \vee (\tau_1^{\max}/\tau)})KL) \wedge 1/(\sqrt{1 \vee (\tau_1^{\max}/\tau)})\sqrt{K\sigma_p^2 dL})$, from Lemma 12, we have

$$\min\{1, C_r/\|\nabla \ell_z(\bar{x}_{k-1})\|\} \leq \min\{1, C_r/\|\nabla \ell_z(x_{k-1}^{(p)})\|\} + \frac{L\|x_{k-1}^{(p)} - \bar{x}_{k-1}\|}{C_r}$$
$$\leq \min\{1, C_r/\|\nabla \ell_z(x_{k-1}^{(p)})\|\} + \widetilde{O}(\eta KL + \eta\sqrt{K}L\sigma_p\sqrt{d})$$
$$\leq \min\{1, C_r/\|\nabla \ell_z(x_{k-1}^{(p)})\|\} + \frac{1}{4}\min\{1, C_r/\|\nabla \ell_z(\bar{x}_{k-1})\|\}$$

Thus, we get $-\min\{1, C_r/\|\nabla \ell_z(x_{k-1}^{(p)})\|\}\ell_z(\bar{x}_{k-1}) \leq -(3/4)\ell_z^{C_r}(\bar{x}_{k-1})$.

In summary, we obtain

$$-\left\langle \bar{x}_{k-1} - x_*, \widehat{\nabla \ell_z^{C_r}}(x_{k-1}^{(p)}) \right\rangle$$
$$\leq -\frac{3}{4}\ell_z^{C_r}(\bar{x}_{k-1}) + \frac{L}{2}\|x_{k-1}^{(p)} - \bar{x}_{k-1}\|^2 - \frac{1}{2\min\{1, C_r/\|\nabla \ell_z(x_{k-1}^{(p)})\|\}L}\|\widehat{\nabla \ell_z^{C_r}}(x_{k-1}^{(p)})\|^2.$$

Averaging this inequality with respect to $z \in D_p$ and $p \in [P]$ results in

$$-\left\langle \bar{x}_{k-1} - x_*, \frac{1}{P}\sum_{p \in [P]} \widehat{\nabla f_p^{C_r}}(x_{k-1}^{(p)}) \right\rangle$$
$$\leq -\frac{3}{4}f^{C_r}(\bar{x}_{k-1}) + \frac{L}{2}\frac{1}{P}\sum_{p \in [P]}\|x_{k-1}^{(p)} - \bar{x}_{k-1}\|^2$$
$$-\frac{1}{2L}\frac{1}{P}\sum_{p \in [P]}\frac{1}{n_p}\sum_{z \in D_p}\frac{1}{\min\{1, C_r/\|\nabla \ell_z(x_{k-1}^{(p)})\|\}}\|\widehat{\nabla \ell_z^{C_r}}(x_{k-1}^{(p)})\|^2.$$

Next, we bound $\left\|\bar{g}_k - \mathbb{E}_{\{I_k^{(p)}\}_{p \in [P]}}[\bar{g}_k]\right\|^2$:

$$\left\|\bar{g}_k - \mathbb{E}_{\{I_k^{(p)}\}_{p \in [P]}}[\bar{g}_k]\right\|^2 \leq \widetilde{O}\left(\frac{C_r^2}{Pb}\right)$$

from Lemma 10.

Since

$$\left\| \frac{1}{P} \sum_{p\in[P]} \widehat{\nabla f_p^{C_r}}(x_{k-1}^{(p)}) \right\|^2 \leq \frac{1}{P} \sum_{p\in[P]} \frac{1}{n_p} \sum_{z\in D_p} \|\widehat{\nabla \ell_z^{C_r}}(x_{k-1}^{(p)})\|^2,$$

combining all the results yields

$$\begin{aligned}
&\|\bar{x}_k - x_*\|^2 \\
&\leq \|\bar{x}_{k-1} - x_*\|^2 - \frac{3\eta}{2} f^{C_r}(\bar{x}_{k-1}) + \eta L \frac{1}{P} \sum_{p\in[P]} \|x_{k-1}^{(p)} - \bar{x}_{k-1}\|^2 \\
&\quad - \frac{\eta}{2L} \frac{1}{P} \sum_{p\in[P]} \frac{1}{n_p} \sum_{z\in D_p} \frac{1}{\min\{1, C_r/\|\nabla \ell_z(x_{k-1}^{(p)})\|\}} \|\widehat{\nabla \ell_z^{C_r}}(x_{k-1}^{(p)})\|^2 \\
&\quad - 2\eta \left\langle \bar{x}_{k-1} - x_*, \bar{g}_k - \mathbb{E}_{\{I_k^{(p)}\}_{p\in[P]}}[\bar{g}_k] \right\rangle - 2\eta \langle \bar{x}_{k-1} - x_*, \bar{\xi}_k \rangle \\
&\quad + \widetilde{O}\left( \frac{\eta^2}{Pb} + \frac{\eta^2 \overline{\sigma^2} d}{P} \right) C_r^2
\end{aligned}$$

under $\eta \leq 1/(4KL)$.

From Lemma 13, we have

$$\frac{1}{P} \sum_{p\in[P]} \|x_{k-1}^{(p)} - \bar{x}_{k-1}\|^2 \leq \frac{1}{P^2} \sum_{p,p'\in[P]} \|x_{k-1}^{(p)} - x_{k-1}^{(p')}\|^2 \leq \widetilde{O}\left( \eta^2 K^2 + \eta^2 K \overline{\sigma^2} d \right) C_r^2$$

and then, we get

$$\begin{aligned}
&\|\bar{x}_k - x_*\|^2 \\
&\leq \|\bar{x}_{k-1} - x_*\|^2 - \Omega(\eta) f^{C_r}(\bar{x}_{k-1}) \\
&\quad + \widetilde{O}\left( \eta^3 K^2 L + \eta^3 K L \overline{\sigma^2} d + \frac{\eta^2 \overline{\sigma^2} d}{P} + \frac{\eta^2}{Pb} \right) C_r^2 \\
&\quad - \Omega\left(\frac{\eta}{L}\right) \frac{1}{P} \sum_{p\in[P]} \frac{1}{n_p} \sum_{z\in D_p} \frac{1}{\min\{1, C_r/\|\nabla \ell_z(x_{k-1}^{(p)})\|\}} \|\widehat{\nabla \ell_z^{C_r}}(x_{k-1}^{(p)})\|^2 \\
&\quad - 2\eta \left\langle \bar{x}_{k-1} - x_*, \bar{g}_k - \mathbb{E}_{\{I_k^{(p)}\}_{p\in[P]}}[\bar{g}_k] \right\rangle - 2\eta \langle \bar{x}_{k-1} - x_*, \bar{\xi}_k \rangle
\end{aligned}$$

Recursively using this inequality and rearranging the result gives

$$\|\bar{x}_K - x_*\|^2$$

$$\leq \|\bar{x}_0 - x_*\|^2 - \Omega(\eta) \sum_{k=1}^{K} \mathbb{E}[f^{C_r}(\bar{x}_{k-1}) \mid E]$$

$$+ \widetilde{O}\left(\eta^3 K^3 L + \eta^3 K^2 L \overline{\sigma^2} d + \frac{\eta^2 K \overline{\sigma^2} d}{P} + \frac{\eta^2 K}{Pb}\right) C_r^2$$

$$- \Omega\left(\frac{\eta}{L}\right) \sum_{k=1}^{K} \frac{1}{P} \sum_{p \in [P]} \frac{1}{n_p} \sum_{z \in D_p} \frac{1}{\min\{1, C_r/\|\nabla \ell_z(x_{k-1}^{(p)})\|\}} \|\widehat{\nabla \ell_z^{C_r}}(x_{k-1}^{(p)})\|^2$$

$$- 2\eta \sum_{k=1}^{K} \left\langle \bar{x}_{k-1} - x_*, \bar{g}_k - \mathbb{E}_{\{I_k^{(p)}\}_{p \in [P]}}[\bar{g}_k] \right\rangle - 2\eta \sum_{k=1}^{K} \langle \bar{x}_{k-1} - x_*, \bar{\xi}_k \rangle.$$

This is the desired result. $\qquad\square$

**Theorem 3** (Convergence Rate). *Suppose that Assumptions 1, 2, 3 and 4 hold. Let $u \in (0,1)$ be sufficiently small. Then, with probability at least $1 - u$, Algorithm 1 with $\widetilde{\nu}$ defined in Proposition 2 and $(\sigma^C, \sigma)$ defined in Theorem 4 satisfies*

$$\frac{1}{KR} \sum_{r=1}^{R} \sum_{k=1}^{K} f^{C_r}(\bar{x}_{k-1,r-1})$$

$$\leq \widetilde{O}\left(\frac{\|x_{\text{ini}} - x_*\|^2 + B^2}{\eta KR} + \frac{\tau \sqrt{R} G^2}{n_{\text{loc}} \sqrt{P} \varepsilon} \left(\eta^2 K^2 L + \frac{\eta KRd}{n_{\text{loc}}^2 P \varepsilon^2} + \frac{\eta}{Pb} + \left(\eta KL + \frac{1}{P}\right) \frac{\eta d}{b^2 \varepsilon}\right)\right).$$

*under $\hat{G} \geq G$ with $\hat{G} = \Theta(G)$, $\tau \leq \tau_1^{\max}$ and*

$$\eta \leq \widetilde{O}\left(\frac{1}{\phi(\tau)KL} \wedge \frac{1}{KGB} \wedge \frac{n_{\text{loc}}\varepsilon}{\phi(\tau)K\sqrt{Rd}L} \wedge \frac{n_{\text{loc}}^2 P \varepsilon^2}{\psi(\tau)KRLd} \wedge \frac{Pb}{\psi(\tau)L}\right),$$

*where $\phi(\tau) := \sqrt{\tau_1^{\max}/\tau} + (\tau_1^{\max}\tau)^{1/4}$, $\psi(\tau) := \sqrt{\tau_1^{\max}\tau}$ and $\tau_q^{\max}$ is defined in Proposition 3.*

*Proof.* From Proposition 3, we have

$$\|x_r - x_*\|^2$$

$$\leq \|x_{r-1} - x_*\|^2 - \Omega(\eta) \sum_{k=1}^{K} \mathbb{E}[f^{C_r}(\bar{x}_{k-1,r-1}) \mid E]$$

$$+ \widetilde{O}\left(\eta^3 K^3 L + \eta^3 K^2 L \overline{\sigma^2} d + \frac{\eta^2 K \overline{\sigma^2} d}{P} + \frac{\eta^2 K}{Pb}\right) C_r^2$$

$$- \Omega\left(\frac{\eta}{L}\right) \sum_{k=1}^{K} \frac{1}{P} \sum_{p \in [P]} \frac{1}{n_p} \sum_{z \in D_p} \frac{1}{\min\{1, C_r/\|\nabla \ell_z(x_{k-1,r-1}^{(p)})\|\}} \|\widehat{\nabla \ell_z^{C_r}}(x_{k-1,r-1}^{(p)})\|^2$$

$$- 2\eta \sum_{k=1}^{K} \left\langle \bar{x}_{k-1,r-1} - x_*, \bar{g}_{k,r-1} - \mathbb{E}_{\{I_{k,r-1}^{(p)}\}_{p \in [P]}}[\bar{g}_{k,r-1}] \right\rangle$$

$$- 2\eta \sum_{k=1}^{K} \langle \bar{x}_{k-1,r-1} - x_*, \bar{\xi}_{k,r-1} \rangle.$$

Here, we used the fact that $\|\text{Clip}(x,B) - x_*\| \leq \|x - x_*\|$ for any $x \in \mathbb{R}^d$ and $\|x_*\| \leq B$ from the elementary projection property.

Recursively using this inequality from $r = 1$ to $R$ yields

$$
\|x_R - x_*\|^2
$$
$$
\leq \|x_0 - x_*\|^2 - \Omega(\eta) \sum_{r=1}^{R} \sum_{k=1}^{K} \mathbb{E}[f^{C_r}(\bar{x}_{k-1,r-1}) \mid E]
$$
$$
+ \widetilde{O}\left(\eta^3 K^3 L + \eta^3 K^2 L \overline{\sigma^2} d + \frac{\eta^2 K \overline{\sigma^2} d}{P} + \frac{\eta^2 K}{Pb}\right) \sum_{r=1}^{R} C_r^2
$$
$$
- \Omega\left(\frac{\eta}{L}\right) \sum_{r=1}^{R} \sum_{k=1}^{K} \frac{1}{P} \sum_{p \in [P]} \frac{1}{n_p} \sum_{z \in D_p} \frac{1}{\min\{1, C_r / \|\nabla \ell_z(x_{k-1,r-1}^{(p)})\|\}} \|\widehat{\nabla \ell_z^{C_r}}(x_{k-1,r-1}^{(p)})\|^2
$$
$$
- 2\eta \sum_{r=1}^{R} \sum_{k=1}^{K} \left\langle \bar{x}_{k-1,r-1} - x_*, \bar{g}_{k,r-1} - \mathbb{E}_{\{I_{k,r-1}^{(p)}\}_{p \in [P]}}[\bar{g}_{k,r-1}] \right\rangle
$$
$$
- 2\eta \sum_{r=1}^{R} \sum_{k=1}^{K} \langle \bar{x}_{k-1,r-1} - x_*, \bar{\xi}_{k,r-1} \rangle.
$$

Observe that $\|\bar{x}_{k-1,r-1} - x_*\|^2 \leq 2\|\bar{x}_{k-1,r-1} - \bar{x}_{0,r-1}\|^2 + 2\|\bar{x}_{0,r-1} - x_*\|^2 \leq \widetilde{O}\left(\eta^2 K^2 C_r^2 + \frac{\eta^2 K^2 \overline{\sigma^2} d C_r^2}{P} + B^2\right) \leq \widetilde{O}\left(\frac{\eta^2 K^2 \overline{\sigma^2} d C_r^2}{P} + B^2\right)$ from Lemma 13 with high probability under $\eta K G \leq B$.

Then, applying Lemma 8 to the third and fourth terms of the right hand side, with probability $1 - 2u$, we get

$$
\left| 2\eta \sum_{r=1}^{R} \sum_{k=1}^{K} \langle \bar{x}_{k-1,r-1} - x_*, \bar{g}_{k,r-1} - \mathbb{E}[\bar{g}_{k,r-1}]\rangle \right| \leq \widetilde{O}\left(\eta \sqrt{\sum_{r=1}^{R} \sum_{k=1}^{K} \left(\frac{\eta^2 K \overline{\sigma^2} d C_r^2}{P} + B^2\right) \frac{C_r^2}{Pb}}\right)
$$
$$
\leq \widetilde{O}\left(\frac{\eta^2 K \sqrt{\overline{\sigma^2} d}}{P\sqrt{b}} + \frac{\eta^2 K}{Pb}\right) \sum_{r=1}^{R} C_r^2 + \widetilde{O}(B^2)
$$
$$
\leq \widetilde{O}\left(\frac{\eta^2 K \overline{\sigma^2} d}{P} + \frac{\eta^2 K}{Pb}\right) \sum_{r=1}^{R} C_r^2 + \widetilde{O}(B^2)
$$

and

$$
\left| 2\eta \sum_{r=1}^{R} \sum_{k=1}^{K} \langle \bar{x}_{k-1,r-1} - x_*, \bar{\xi}_{k,r-1} \rangle \right| \leq \widetilde{O}\left(2\eta \sqrt{\sum_{r=1}^{R} \sum_{k=1}^{K} \left(\frac{\eta^2 K \overline{\sigma^2} d C_r^2}{P} + B^2\right) \frac{\overline{\sigma^2} C_r^2 d}{P}}\right)
$$
$$
\leq \widetilde{O}\left(\frac{\eta^2 K \overline{\sigma^2} d}{P}\right) \sum_{r=1}^{R} C_r^2 + \widetilde{O}(B^2).
$$

Thus, we have

$$\|x_R - x_*\|^2$$

$$\leq \|x_0 - x_*\|^2 + \widetilde{O}(B^2) - \Omega(\eta) \sum_{r=1}^{R} \sum_{k=1}^{K} \mathbb{E}[f^{C_r}(\bar{x}_{k-1,r-1}) \mid E]$$

$$+ \widetilde{O}\left(\eta^3 K^3 L + \eta^3 K^2 L \overline{\sigma^2} d + \frac{\eta^2 K \overline{\sigma^2} d}{P} + \frac{\eta^2 K}{Pb}\right) \sum_{r=1}^{R} C_r^2$$

$$- \Omega\left(\frac{\eta}{L}\right) \sum_{r=1}^{R} \sum_{k=1}^{K} \frac{1}{P} \sum_{p \in [P]} \frac{1}{n_p} \sum_{z \in D_p} \frac{1}{\min\{1, C_r/\|\nabla \ell_z(x_{k-1,r-1}^{(p)})\|\}} \|\widehat{\nabla \ell_z^{C_r}}(x_{k-1,r-1}^{(p)})\|^2.$$

From the definition of $C_r$ and Proposition 2, for each round $r$ it holds that

$$C_r^2 = \frac{2\tau}{P} \sum_{p \in [P]} G_r^{(p),C} + O(\tau\widetilde{\nu})$$

$$= \frac{2\tau}{P} \sum_{p \in [P]} \frac{1}{n_p} \sum_{z \in D_p} \|\nabla \ell_z(x_{r-1})\|^2 + O(\tau\widetilde{\nu})$$

$$\leq \frac{2\tau\sqrt{1 \vee \frac{\tau_1(x_{r-1})}{\tau}}}{P} \sum_{p \in [P]} \frac{1}{n_p} \sum_{z \in D_p} \min\{1, C_r/\|\nabla \ell_z(x_{r-1})\|\}\|\nabla \ell_z(x_{r-1})\|^2 + O(\tau\widetilde{\nu}).$$

Here, for the inequality, we used Proposition 2.

Since $\tau \leq \tau_1^{\max}$, choosing appropriate $\eta \leq \widetilde{O}(1/(\sqrt{\tau_1^{\max}/\tau}KL) \wedge 1/(\sqrt{\tau_1^{\max}/\tau}\sqrt{K\sigma_p^2 d}L))$, from Lemma 12, we have

$$\min\{1, C_r/\|\nabla \ell_z(\bar{x}_{k-1})\|\} \leq \min\{1, C_r/\|\nabla \ell_z(x_{k-1}^{(p)})\|\} + \frac{L\|x_{k-1}^{(p)} - \bar{x}_{k-1}\|}{C_r}$$

$$\leq \min\{1, C_r/\|\nabla \ell_z(x_{k-1}^{(p)})\|\} + \widetilde{O}(\eta KL + \eta\sqrt{K}L\sigma_p\sqrt{d})$$

$$\leq \min\{1, C_r/\|\nabla \ell_z(x_{k-1}^{(p)})\|\} + \frac{1}{4}\min\{1, C_r/\|\nabla \ell_z(\bar{x}_{k-1})\|\}$$

Observe that

$$\min\{1, C_r/\|\nabla \ell_z(x_{r-1})\|\}\|\nabla \ell_z(x_{r-1})\|^2$$

$$\leq 2\min\{1, C_r/\|\nabla \ell_z(x_{r-1})\|\}\|\nabla \ell_z(x_{k-1,r-1}^{(p)})\|^2 + 2\|\nabla \ell_z(x_{k-1,r-1}^{(p)}) - \nabla \ell_z(x_{r-1})\|^2$$

$$\leq O(\min\{1, C_r/\|\nabla \ell_z(x_{k-1,r-1}^{(p)})\|\}\|\nabla \ell_z(x_{k-1,r-1}^{(p)})\|^2 + L^2\|x_{k-1,r-1}^{(p)} - x_{0,r-1}^{(p)}\|^2)$$

$$\leq O\left(\frac{1}{\min\{1, C_r/\|\nabla \ell_z(x_{k-1,r-1}^{(p)})\|\}} \|\widehat{\nabla \ell_z^{C_r}}(x_{k-1,r-1}^{(p)})\|^2 + (\eta^2 K^2 L^2 + \eta^2 KL^2\sigma_p^2 d)C_r^2\right).$$

Here, for the second inequality, we used Assumption 4. Furthermore, if we appropriately choose $\tau \leq \tau_1^{\max}$ and $\eta \leq \widetilde{O}(1/((\tau_1^{\max}\tau)^{1/4}KL) \wedge 1/((\tau_1^{\max}\tau)^{1/4}\sqrt{K\sigma_p^2 d}L))$, we get

$$C_r^2 \leq O\left(\frac{\sqrt{\tau_1^{\max}\tau}}{K}\right) \sum_{k=1}^{K} \frac{1}{P} \sum_{p \in [P]} \frac{1}{n_p} \sum_{z \in D_p} \frac{1}{\min\{1, C_r/\|\nabla \ell_z(x_{k-1,r-1}^{(p)})\|\}} \|\widehat{\nabla \ell_z^{C_r}}(x_{k-1,r-1}^{(p)})\|^2 + O(\tau\widetilde{\nu}).$$

Let

$$A := \eta^3 K^3 L + \eta^3 K^2 L \overline{\sigma^2} d + \frac{\eta^2 K \overline{\sigma^2} d}{P} + \frac{\eta^2 K}{Pb}$$

Then, if we appropriately choose $\eta$ satisfying

$$\widetilde{O}\left(A \frac{\sqrt{\tau_1^{\max} \tau}}{K}\right) \leq \Omega\left(\frac{\eta}{L}\right)$$

to cancel out the term $\widetilde{O}(AC_r^2)$, from Proposition 3, we obtain

$$\|x_R - x_*\|^2 \leq \|x_0 - x_*\|^2 + \widetilde{O}(B^2) + \widetilde{O}\left(R\tau\widetilde{\nu}A\right) - \Omega(\eta) \sum_{r=1}^R \sum_{k=1}^K \mathbb{E}[f^{C_r}(\bar{x}_{k-1,r-1}) \mid E],$$

which implies

$$\frac{1}{KR} \sum_{r=1}^R \sum_{k=1}^K f^{C_r}(\bar{x}_{k-1,r-1})$$

$$\leq \widetilde{O}\left(\frac{\|x_{\mathrm{ini}} - x_*\|^2 + B^2}{\eta KR} + \frac{\tau\widetilde{\nu}A}{\eta K}\right)$$

$$= \widetilde{O}\left(\frac{\|x_{\mathrm{ini}} - x_*\|^2 + B^2}{\eta KR} + \frac{\tau\sqrt{\overline{(\sigma^C)^2}}G^2}{\sqrt{P}}\left(\eta^2 K^2 L + \eta^2 KL\overline{\sigma^2}d + \frac{\eta\overline{\sigma^2}d}{P} + \frac{\eta}{Pb}\right)\right)$$

$$= \widetilde{O}\left(\frac{\|x_{\mathrm{ini}} - x_*\|^2 + B^2}{\eta KR} + \frac{\tau\sqrt{R}G^2}{n_{\mathrm{loc}}\sqrt{P}\varepsilon}\left(\eta^2 K^2 L + \frac{\eta KRd}{n_{\mathrm{loc}}^2 P\varepsilon^2} + \frac{\eta}{Pb} + \left(\eta^2 KLd + \frac{\eta d}{P}\right)\frac{1}{b^2\varepsilon}\right)\right).$$

Here, we used

$$\overline{(\sigma^C)^2} = \widetilde{O}\left(\frac{R}{(n_{\mathrm{loc}}\varepsilon)^2}\right)$$

and

$$\overline{\sigma^2} = \widetilde{O}\left(\frac{KR}{(n_{\mathrm{loc}}\varepsilon)^2} + \frac{1}{b^2\varepsilon}\right).$$

Finally, we summarize a sufficient condition on $\eta$:

$$\eta \leq \widetilde{O}\left(\frac{1}{\phi(\tau)KL} \wedge \frac{1}{KGB} \wedge \frac{n_{\mathrm{loc}}\varepsilon}{\phi(\tau)K\sqrt{Rd}L} \wedge \frac{n_{\mathrm{loc}}^2 P\varepsilon^2}{\psi(\tau)KRLd} \wedge \frac{Pb}{\psi(\tau)L}\right),$$

where $\phi(\tau) := \sqrt{\tau_1^{\max}/\tau} + (\tau_1^{\max}\tau)^{1/4}$ and $\psi(\tau) := \sqrt{\tau_1^{\max}\tau}$.

This finishes the proof. $\qquad\square$

**Corollary 2.** *It is assumed that $\|x_{\mathrm{ini}} - x_*\|$, $B$, $G$ and $L$ are $\Theta(1)$. Then, for any $q \in [0.5, 1.0]$, there exists $D_q \subset D$ such that $|D_q| \geq q|D|$ and Algorithm 1 with $(\varepsilon, \delta)$-DP guarantees satisfies*

$$f_{D^q}\left(x^{(\mathrm{out})}\right)$$

$$\leq \widetilde{O}\left(\sqrt{1 \vee \frac{\tau_q^{\max}}{\tau}}\left(\frac{\tau^{\frac{4}{9}}}{d^{\frac{2}{9}}}(\epsilon_{\mathrm{util}}^{DP\text{-}SGD})^{\frac{10}{9}} + \frac{(\phi(\tau)\tau\sqrt{P})^{\frac{1}{3}}}{d^{\frac{1}{6}}}(\epsilon_{\mathrm{util}}^{DP\text{-}SGD})^{\frac{4}{3}} + \frac{\sqrt{\tau}}{d^{\frac{1}{4}}}(\epsilon_{\mathrm{util}}^{DP\text{-}SGD})^{\frac{3}{2}} + \frac{\tau^{\frac{7}{15}}}{d^{\frac{7}{30}}}(\epsilon_{\mathrm{util}}^{DP\text{-}SGD})^{\frac{19}{15}}\right.\right.$$

$$\left.\left. + \frac{(\phi(\tau)\tau^2\sqrt{P})^{\frac{1}{5}}}{d^{\frac{1}{5}}}(\epsilon_{\mathrm{util}}^{DP\text{-}SGD})^{\frac{7}{5}} + \frac{\psi(\tau)^{\frac{1}{5}}}{d^{\frac{1}{5}}}(\epsilon_{\mathrm{util}}^{DP\text{-}SGD})^{\frac{8}{5}} + \frac{\tau^{\frac{2}{5}}}{(\phi(\tau)d)^{\frac{1}{5}}}(\epsilon_{\mathrm{util}}^{DP\text{-}SGD})^{\frac{6}{5}} + \psi(\tau)(\epsilon_{\mathrm{util}}^{DP\text{-}SGD})^2\right)\right).$$

*under the conditions in Theorem 2, where $f_{D^q}(x) := (1/|D^q|)\sum_{z \in D^q} \ell_z(x)$ and $\epsilon_{\mathrm{util}}^{DP\text{-}SGD} = \widetilde{\Theta}(\sqrt{d}/(n_{\mathrm{loc}}\sqrt{P}\varepsilon))$.*

*Proof.* Let $q \in [0.5, 1]$. First, note that

$$D^q := \left\{ z \in D : \forall r \in [R], \forall k \in [K] : \|\nabla \ell_z(\bar{x}_{k-1, r-1})\| \leq \sqrt{1 \vee \frac{\tau_q(x_{r-1})}{\tau}} C_r \right\}$$

satisfies $|D^q| \geq q|D|$ from Proposition 2..

Hence, when we define

$$f_{D^q}(x) := \frac{1}{|D^q|} \sum_{z \in D^q} \ell_z(x),$$

from Theorem 2 we have

$$f_{D^q}\left(x^{(\text{out})}\right)$$

$$\leq \frac{1}{R} \sum_{r=1}^{R} \frac{1}{K} \sum_{k=1}^{K} f_{D^q}(\bar{x}_{k-1, r-1})$$

$$\leq O\left( \frac{\sqrt{1 \vee \frac{\tau_q^{\max}}{\tau}}}{R} \sum_{r=1}^{R} \frac{1}{K} \sum_{k=1}^{K} f^{C_r}(\bar{x}_{k-1, r-1}) \right)$$

$$\leq \tilde{O}\left( \sqrt{1 \vee \frac{\tau_q^{\max}}{\tau}} \left( \frac{\|x_{\text{ini}} - x_*\|^2 + B^2}{\eta K R} + \frac{\tau \sqrt{R} G^2}{n_{\text{loc}} \sqrt{P} \varepsilon} \left( \eta^2 K^2 L + \frac{\eta K R d}{n_{\text{loc}}^2 P \varepsilon^2} + \frac{\eta}{Pb} + \left( \eta K L + \frac{1}{P} \right) \frac{\eta d}{b^2 \varepsilon} \right) \right) \right)$$

$$= \tilde{O}\left( \sqrt{1 \vee \frac{\tau_q^{\max}}{\tau}} \left( \frac{\|x_{\text{ini}} - x_*\|^2 + B^2}{\eta K R} + \frac{\tau \eta K R^{\frac{3}{2}} G^2 d}{(n_{\text{loc}} \sqrt{P} \varepsilon)^3} + \frac{\tau \sqrt{R} G^2}{n_{\text{loc}} \sqrt{P} \varepsilon} \left( \eta^2 K^2 L + \frac{\eta}{Pb} + \left( \eta K L + \frac{1}{P} \right) \frac{\eta d}{b^2 \varepsilon} \right) \right) \right)$$

$$= \tilde{O}\left( \sqrt{1 \vee \frac{\tau_q^{\max}}{\tau}} \left( \frac{1}{\eta K R} + \frac{\tau \eta K R^{\frac{3}{2}} d}{(n_{\text{loc}} \sqrt{P} \varepsilon)^3} + \frac{\tau \sqrt{R}}{n_{\text{loc}} \sqrt{P} \varepsilon} \left( \eta^2 K^2 + \frac{\eta}{Pb} + \left( \eta K + \frac{1}{P} \right) \frac{\eta d}{b^2 \varepsilon} \right) \right) \right).$$

Here, for the first inequality, we used convexity of $f_{D^q}$. The second inequality holds because

$$f_{D^q}(\bar{x}_{k-1, r-1}) \leq \frac{2}{n} \sum_{z \in D^q} \ell_z(\bar{x}_{k-1, r-1})$$

$$\leq \frac{2\sqrt{1 \vee \frac{\tau_q(x_{r-1})}{\tau}}}{n} \sum_{z \in D^q} \min\{1, C_r / \|\nabla \ell_z(\bar{x}_{k-1, r-1})\|\} \ell_z(\bar{x}_{k-1, r-1})$$

$$\leq \frac{2\sqrt{1 \vee \frac{\tau_q(x_{r-1})}{\tau}}}{n} \sum_{z \in D} \min\{1, C_r / \|\nabla \ell_z(\bar{x}_{k-1, r-1})\|\} \ell_z(\bar{x}_{k-1, r-1})$$

$$\leq O\left( \sqrt{1 \vee \frac{\tau_q(x_{r-1})}{\tau}} f^{C_r}(\bar{x}_{k-1, r-1}) \right).$$

To derive the best achievable utility, we need to determine the optimal choices of $\eta$, $K$, and $R$. We first determine $R$.

- $R \leq \tilde{O}\left( \frac{(n_{\text{loc}} \sqrt{P} \varepsilon)^{\frac{6}{5}}}{(\tau d)^{\frac{2}{5}} (\eta K)^{\frac{4}{5}}} \right) =: R_1$ is a sufficient condition for 1st term $\geq$ 2nd term

- $R \leq \tilde{O}\left( \frac{(n_{\text{loc}} \sqrt{P} \varepsilon)^{\frac{2}{3}}}{\tau^{\frac{2}{3}} (\eta K)^2 + \tau^{\frac{2}{3}} \eta^{\frac{2}{3}} (\eta K)^{\frac{2}{3}} \left( \frac{1}{Pb} + (\eta K + \frac{1}{P}) \frac{d}{b^2 \varepsilon} \right)} \right) =: R_2$ is a sufficient condition for 1st term $\geq$ 3rd term,

Also, from the condition on $\eta$ in Theorem 2, we need

- $R \le \widetilde{O}\left(\frac{(n_{\text{loc}}\varepsilon)^2}{\phi(\tau)^2(\eta K)^2 d}\right) =: R_3,$

- $R \le \widetilde{O}\left(\frac{(n_{\text{loc}}\sqrt{P}\varepsilon)^2}{\psi(\tau)\eta K d}\right) =: R_4.$

Thus, if $R_1, R_2, R_3, R_4 \ge 1$, setting $R_* := R_1 \wedge R_2 \wedge R_3 \wedge R_4$ gives

$$
\begin{aligned}
&f_{D^q}\left(x^{(\text{out})}\right) \\
&\le \widetilde{O}\left(\sqrt{1 \vee \frac{\tau_q^{\max}}{\tau}}\left(\frac{1}{\eta K R_*}\right)\right) \\
&\le \widetilde{O}\left(\sqrt{1 \vee \frac{\tau_q^{\max}}{\tau}}\left(\frac{(\tau d)^{\frac{2}{5}}}{(\eta K)^{\frac{1}{5}}(n_{\text{loc}}\sqrt{P}\varepsilon)^{\frac{6}{5}}} + \frac{\tau^{\frac{2}{3}}\eta K}{(n_{\text{loc}}\sqrt{P}\varepsilon)^{\frac{2}{3}}} + \frac{\tau^{\frac{2}{3}}\eta^{\frac{1}{3}}}{K^{\frac{1}{3}}(n_{\text{loc}}\sqrt{P}\varepsilon)^{\frac{2}{3}}}\left(\frac{1}{Pb} + \left(\eta K + \frac{1}{P}\right)\frac{d}{b^2\varepsilon}\right)\right.\right. \\
&\qquad \left.\left. + \frac{\psi(\tau)d}{(n_{\text{loc}}\sqrt{P}\varepsilon)^2} + \frac{\phi(\tau)^2\eta K d}{(n_{\text{loc}}\varepsilon)^2}\right)\right).
\end{aligned}
$$

Note that 3rd term can be arbitrary small by reducing $\eta$ and increasing $K$ for any fixed optimal choice of $(\eta K)_*$ when we focus on the asymptotic utility bound. Thus, we ignore this term.

Now, we need to determine the optimal choice of $\eta K$.

- $\eta K \le \widetilde{O}\left(\frac{d^{\frac{1}{3}}}{\tau^{\frac{2}{9}}(n_{\text{loc}}\sqrt{P}\varepsilon)^{\frac{4}{9}}}\right) =: (\eta K)_1$ is a sufficient condition for 1st term $\ge$ 2nd term,

- $\eta K \le \widetilde{O}\left(\frac{\tau^{\frac{1}{3}}(n_{\text{loc}}\varepsilon)^{\frac{2}{3}}}{\phi(\tau)^{\frac{5}{3}}\sqrt{P}d}\right) =: (\eta K)_2$ is a sufficient condition for 1st term $\ge$ 5th term,

- $\eta K \le \widetilde{O}\left(\frac{(n_{\text{loc}}\sqrt{P}\varepsilon)^{\frac{3}{2}}}{\sqrt{\tau d}}\right) =: (\eta K)_3$ is a sufficient condition for $R_1 \ge 1$ with sufficiently small $\eta$ and large $K$,

- $\eta K \le \widetilde{O}\left(\frac{(n_{\text{loc}}\sqrt{P}\varepsilon)^{\frac{1}{3}}}{\tau^{\frac{1}{3}}}\right) =: (\eta K)_4$ is a sufficient condition for $R_2 \ge 1$,

- $\eta K \le \widetilde{O}\left(\frac{n_{\text{loc}}\varepsilon}{\phi(\tau)\sqrt{d}}\right) =: (\eta K)_5$ is a sufficient condition for $R_3 \ge 1$,

- $\eta K \le \widetilde{O}\left(\frac{(n_{\text{loc}}\sqrt{P}\varepsilon)^2}{\psi(\tau)d}\right) =: (\eta K)_6$ is a sufficient condition for $R_4 \ge 1$.

Also, from the condition on $\eta$ in Theorem 2, we need

$$
\eta K \le \widetilde{O}\left(\frac{1}{\phi(\tau)} \wedge \frac{KPb}{\psi(\tau)}\right) := (\eta K)_7.
$$

Thus, setting $(\eta K)_* := (\eta K)_1 \wedge (\eta K)_2 \wedge (\eta K)_3 \wedge (\eta K)_4 \wedge (\eta K)_5 \wedge (\eta K)_6 \wedge (\eta K)_7$ yields an asymptotic utility bound of

$$f_{D^q}\left(x^{(\text{out})}\right)$$

$$\leq \widetilde{O}\left(\sqrt{1 \vee \frac{\tau_q^{\max}}{\tau}}\left(\frac{(\tau d)^{\frac{2}{5}}}{(\eta K)_*^{\frac{1}{5}}(n_{\text{loc}}\sqrt{P}\varepsilon)^{\frac{6}{5}}} + \frac{\psi(\tau)d}{(n_{\text{loc}}\sqrt{P}\varepsilon)^2}\right)\right)$$

$$\leq \widetilde{O}\left(\sqrt{1 \vee \frac{\tau_q^{\max}}{\tau}}\left(\frac{(\tau d)^{\frac{2}{5}}}{(\eta K)_*^{\frac{1}{5}}(n_{\text{loc}}\sqrt{P}\varepsilon)^{\frac{6}{5}}} + \frac{\psi(\tau)d}{(n_{\text{loc}}\sqrt{P}\varepsilon)^2}\right)\right)$$

$$\leq \widetilde{O}\left(\sqrt{1 \vee \frac{\tau_q^{\max}}{\tau}}\left(\frac{\tau^{\frac{4}{9}}d^{\frac{1}{3}}}{(n_{\text{loc}}\sqrt{P}\varepsilon)^{\frac{10}{9}}} + \frac{\phi(\tau)^{\frac{1}{3}}\tau^{\frac{1}{3}}\sqrt{d}}{(n_{\text{loc}}\varepsilon)^{\frac{1}{3}}(n_{\text{loc}}\sqrt{P}\varepsilon)} + \frac{\sqrt{\tau}d}{(n_{\text{loc}}\sqrt{P}\varepsilon)^{\frac{3}{2}}} + \frac{\tau^{\frac{7}{15}}d^{\frac{2}{5}}}{(n_{\text{loc}}\sqrt{P}\varepsilon)^{\frac{19}{15}}}\right.\right.$$

$$\left.\left. + \frac{\phi(\tau)^{\frac{1}{5}}\tau^{\frac{2}{5}}\sqrt{d}}{(n_{\text{loc}}\varepsilon)^{\frac{1}{5}}(n_{\text{loc}}\sqrt{P}\varepsilon)^{\frac{6}{5}}} + \frac{\psi(\tau)^{\frac{1}{5}}d^{\frac{3}{5}}}{(n_{\text{loc}}\sqrt{P}\varepsilon)^{\frac{8}{5}}} + \frac{(\tau d)^{\frac{2}{5}}}{\phi(\tau)^{\frac{1}{5}}(n_{\text{loc}}\sqrt{P}\varepsilon)^{\frac{6}{5}}} + \frac{\psi(\tau)d}{(n_{\text{loc}}\sqrt{P}\varepsilon)^2}\right)\right)$$

$$= \widetilde{O}\left(\sqrt{1 \vee \frac{\tau_q^{\max}}{\tau}}\left(\frac{\tau^{\frac{4}{9}}}{d^{\frac{2}{9}}}(\epsilon_{\text{util}}^{\text{DP-SGD}})^{\frac{10}{9}} + \frac{(\phi(\tau)\tau\sqrt{P})^{\frac{1}{3}}}{d^{\frac{1}{6}}}(\epsilon_{\text{util}}^{\text{DP-SGD}})^{\frac{4}{3}} + \frac{\sqrt{\tau}}{d^{\frac{1}{4}}}(\epsilon_{\text{util}}^{\text{DP-SGD}})^{\frac{3}{2}} + \frac{\tau^{\frac{7}{15}}}{d^{\frac{7}{30}}}(\epsilon_{\text{util}}^{\text{DP-SGD}})^{\frac{19}{15}}\right.\right.$$

$$\left.\left. + \frac{(\phi(\tau)\tau^2\sqrt{P})^{\frac{1}{5}}}{d^{\frac{1}{5}}}(\epsilon_{\text{util}}^{\text{DP-SGD}})^{\frac{7}{5}} + \frac{\psi(\tau)^{\frac{1}{5}}}{d^{\frac{1}{5}}}(\epsilon_{\text{util}}^{\text{DP-SGD}})^{\frac{8}{5}} + \frac{\tau^{\frac{2}{5}}}{(\phi(\tau)d)^{\frac{1}{5}}}(\epsilon_{\text{util}}^{\text{DP-SGD}})^{\frac{6}{5}} + \psi(\tau)(\epsilon_{\text{util}}^{\text{DP-SGD}})^2\right)\right).$$

This finishes the proof. □

## D   DP-FedAvg

Here, we describe the implementation details of DP-FedAvg, which is a natural baseline in our experiments.

---

**Algorithm 2** DP-FedAvg
---
1: **for** $r = 1$ to $R$ **do**
2:    **for** $p \in [P]$ in parallel **do**
3:       Compute $\sigma_p$ based on (3).
4:       $x_{0,r-1}^{(p)} = x_{r-1}$
5:       **for** $k = 1$ to $K$ **do**
6:          Sample minibatch $I_{k,r-1}^{(p)} \sim D_p$ with size $b$.
7:          $g_{k,r-1}^{(p)} = \frac{1}{b}\sum_{z \in I_{k,r-1}^{(p)}} \text{Clip}(\nabla\ell_z(x_{k-1,r-1}^{(p)}), C)$
8:          $x_{k,r-1}^{(p)} = x_{k-1,r-1}^{(p)} - \eta(g_{k,r-1}^{(p)} + \xi_{k,r-1}^{(p)})$, where $\xi_{k,r-1}^{(p)} \sim N(0, \sigma_p^2 C^2 I)$.
9:       **end for**
10:    **end for**
11:    $x_r = \frac{1}{P}\sum_{p \in [P]} x_{K,r-1}^{(p)}$.
12: **end for**
13: **Return:** $x_R$.
---

The concrete procedures of DP-FedAvg are given in Algorithm 2 for record level centralized DP.

As in Section 4, we assume that the minibatch sampling is random permutation. The following proposition and theorems are natural counter parts of Proposition 6 and Theorem 4 respectively. The proofs are completely same as the ones of Proposition 6 and Theorem 4, and we omit them.

**Proposition 6** (One-round RDP analysis). *Suppose that $b \leq n_p$ for $p \in [P]$. Let $\Delta := 2/b$ and $\gamma_p := b/n_p$. We define $\varepsilon_p(\alpha) := \alpha\Delta^2/(2\sigma_p^2)$. Given $x_{r-1}$, $\{x_{k,r-1}^{(p)}\}_{k \in [K]}$ is $(\alpha, K\mathcal{S}(\varepsilon_p(\cdot), \gamma_p)(\alpha))$-RDP for any integer $\alpha \geq 2$.*

**Theorem 4** (DP noise size)**.** *Let $\alpha_* := 1 + \lceil 2\log(1/\delta)/\varepsilon \rceil$. Then, the output information of client $p \in [P]$ in Algorithm 1 is $(\varepsilon, \delta)$-DP, if*

$$\sigma_p := \min\{\sigma_p \geq 0 \mid RKS(\varepsilon_p(\cdot), \gamma_p)(\alpha_*) \leq \varepsilon_{\text{target}}\} \tag{3}$$

*with $\varepsilon_{\text{target}} := \varepsilon/2$. In particular, the DP noise size $\sigma_p$ in (1) satisfies*

$$\sigma_p = O\left(\frac{\sqrt{KR\log\frac{1}{\delta}}}{n_p\varepsilon} \vee \frac{\sqrt{\log\frac{1}{\delta}}}{b\sqrt{\varepsilon}}\right)$$

*under $b \leq \frac{n_p}{2e\alpha_*} \wedge \left(\frac{4n_p}{\alpha_*\sigma_p^2}\right)^{\frac{1}{3}}$ and $\varepsilon = O(\log(1/\delta))$.*

# E    Additional experiments

## E.1    Hyperparameter Tuning

We briefly describe the hyperparameter tuning process for both the conventional DP-FedAvg and the proposed AdaptDP-FedAvg. Both methods require setting the clipping radii $C, \hat{G}$, as well as the learning rate $\eta$, prior to model training. These optimal values generally depend on the target privacy level and task. Therefore, we performed an empirical search to identify the hyperparameters that minimize the training loss for each dataset, using the three privacy levels listed in Table 1. The results of this hyperparameter search are presented in Table 3. For each privacy level and task in Table 1, the configuration that achieved the lowest training loss is used to report the final results in Table 2.

Table 3:   Training loss for each hyperparameter setting tested.

| Dataset | Hyperparameter | | DP-FedAvg | | | AdaptDP-FedAvg | | |
|---|---|---|---|---|---|---|---|---|
| | $\eta$ | $C$ or $\hat{G}$ | Lv.1 | Lv.2 | Lv.3 | Lv.1 | Lv.2 | Lv.3 |
| T1) Artificial dataset | 0.1 | 0.5 | $3.88e^{-0}$ | $3.89e^{-0}$ | $3.91e^{-0}$ | $3.88e^{-0}$ | $3.92e^{-0}$ | $3.93e^{-0}$ |
| $(R,K)=(150,20)$ | 0.1 | 1 | $8.11e^{-6}$ | $3.24e^{-5}$ | $9.01e^{-5}$ | $\mathbf{3.34e^{-7}}$ | $4.72e^{-6}$ | $2.34e^{-5}$ |
| | 0.1 | 3 | $6.77e^{-5}$ | $2.71e^{-4}$ | $7.53e^{-4}$ | $5.02e^{-7}$ | $\mathbf{3.81e^{-6}}$ | $\mathbf{1.82e^{-5}}$ |
| | 0.1 | 5 | $1.88e^{-4}$ | $7.52e^{-4}$ | $2.09e^{-3}$ | $2.48e^{-6}$ | $1.82e^{-5}$ | $5.30e^{-5}$ |
| | 0.3 | 0.5 | $\mathbf{6.73e^{-6}}$ | $\mathbf{2.69e^{-5}}$ | $\mathbf{7.62e^{-5}}$ | $4.29e^{-6}$ | $9.13e^{-5}$ | $1.14e^{-4}$ |
| | 0.3 | 1 | $2.80e^{-5}$ | $1.12e^{-4}$ | $3.15e^{-4}$ | $3.96e^{-6}$ | $1.69e^{-4}$ | $2.56e^{-4}$ |
| | 0.3 | 3 | $2.52e^{-4}$ | $1.01e^{-3}$ | $2.83e^{-3}$ | $3.69e^{-6}$ | $1.66e^{-4}$ | $2.49e^{-4}$ |
| | 0.3 | 5 | $2.80e^{-3}$ | $2.80e^{-3}$ | $7.86e^{-3}$ | $3.94e^{-6}$ | $1.61e^{-4}$ | $2.40e^{-4}$ |
| | 0.5 | 0.5 | $1.36e^{-5}$ | $5.44e^{-5}$ | $1.61e^{-4}$ | $1.37e^{-7}$ | $1.77e^{-4}$ | $2.27e^{-4}$ |
| | 0.5 | 1 | $5.08e^{-5}$ | $2.04e^{-4}$ | $6.19e^{-4}$ | $1.36e^{-5}$ | $6.20e^{-4}$ | $8.91e^{-4}$ |
| | 0.5 | 3 | $4.57e^{-4}$ | $1.84e^{-3}$ | $5.59e^{-3}$ | $1.18e^{-5}$ | $6.80e^{-4}$ | $1.09e^{-3}$ |
| | 0.5 | 5 | $1.27e^{-3}$ | $5.10e^{-3}$ | $1.55e^{-2}$ | $7.85e^{-6}$ | $5.99e^{-4}$ | $8.85e^{-4}$ |
| T2) Bank Marketing | 0.01 | 0.1 | $3.15e^{-1}$ | $3.16e^{-1}$ | $3.14e^{-1}$ | $3.15e^{-1}$ | $3.17e^{-1}$ | $3.14e^{-1}$ |
| $(R,K)=(100,226)$ | 0.01 | 0.5 | $3.14e^{-1}$ | $3.16e^{-1}$ | $3.14e^{-1}$ | $3.14e^{-1}$ | $3.16e^{-1}$ | $3.14e^{-1}$ |
| | 0.03 | 0.05 | $\mathbf{3.13e^{-1}}$ | $3.14e^{-1}$ | $3.14e^{-1}$ | $\mathbf{3.12e^{-1}}$ | $3.13e^{-1}$ | $3.14e^{-1}$ |
| | 0.03 | 0.1 | $3.21e^{-1}$ | $3.13e^{-1}$ | $3.13e^{-1}$ | $3.15e^{-1}$ | $\mathbf{3.12e^{-1}}$ | $\mathbf{3.13e^{-1}}$ |
| | 0.03 | 0.5 | $3.15e^{-1}$ | $\mathbf{3.12e^{-1}}$ | $3.14e^{-1}$ | $3.15e^{-1}$ | $3.14e^{-1}$ | $3.14e^{-1}$ |
| | 0.1 | 0.05 | $3.15e^{-1}$ | $3.14e^{-1}$ | $3.14e^{-1}$ | $3.14e^{-1}$ | $3.13e^{-1}$ | $3.15e^{-1}$ |
| | 0.1 | 0.1 | $3.15e^{-1}$ | $3.14e^{-1}$ | $\mathbf{3.13e^{-1}}$ | $3.15e^{-1}$ | $3.12e^{-1}$ | $3.16e^{-1}$ |
| | 0.1 | 0.5 | $3.16e^{-1}$ | $3.14e^{-1}$ | $3.15e^{-1}$ | $3.15e^{-1}$ | $3.12e^{-1}$ | $3.16e^{-1}$ |
| T3) MNIST | 0.1 | 0.05 | $6.91e^{-4}$ | $5.96e^{-4}$ | $5.62e^{-4}$ | $4.49e^{-4}$ | $3.49e^{-4}$ | $5.70e^{-4}$ |
| $(R,K)=(100,15)$ | 0.1 | 0.1 | $1.54e^{-1}$ | $2.86e^{-4}$ | $\mathbf{4.06e^{-4}}$ | $3.27e^{-4}$ | $2.22e^{-4}$ | $2.51e^{-4}$ |
| | 0.1 | 0.5 | $7.80e^{-2}$ | $7.98e^{-5}$ | $1.28e^{-1}$ | $2.63e^{-4}$ | $2.03e^{-4}$ | $9.16e^{-5}$ |
| | 0.3 | 0.05 | $\mathbf{4.84e^{-5}}$ | $\mathbf{5.57e^{-5}}$ | $7.79e^{-4}$ | $\mathbf{4.25e^{-5}}$ | $\mathbf{3.38e^{-5}}$ | $\mathbf{4.02e^{-5}}$ |
| | 0.3 | 0.1 | $4.57e^{-2}$ | $3.10e^{-2}$ | $9.31e^{-2}$ | $4.30e^{-5}$ | $3.97e^{-5}$ | $4.96e^{-5}$ |
| | 0.3 | 0.5 | $2.97e^{-3}$ | $4.40e^{-4}$ | $6.32e^{-2}$ | $2.04e^{-4}$ | $4.04e^{-5}$ | $8.36e^{-5}$ |
| T4) FashionMNIST | 0.03 | 0.05 | $1.58e^{-1}$ | $1.30e^{-1}$ | $1.45e^{-1}$ | $1.25e^{-1}$ | $1.25e^{-1}$ | $1.37e^{-1}$ |
| $(R,K)=(100,15)$ | 0.03 | 0.1 | $1.38e^{-1}$ | $1.05e^{-1}$ | $\mathbf{1.28e^{-1}}$ | $9.65e^{-2}$ | $1.09e^{-1}$ | $1.45e^{-1}$ |
| | 0.03 | 0.5 | $2.06e^{-2}$ | $\mathbf{3.39e^{-2}}$ | $8.64e^{-1}$ | $1.03e^{-2}$ | $1.97e^{-2}$ | $7.84e^{-1}$ |
| | 0.1 | 0.05 | $2.39e^{-1}$ | $1.23e^{-1}$ | $1.52e^{-1}$ | $1.62e^{-1}$ | $1.04e^{-1}$ | $2.04e^{-1}$ |
| | 0.1 | 0.1 | $1.97e^{-1}$ | $4.82e^{-2}$ | $1.60e^{-1}$ | $1.54e^{-1}$ | $8.07e^{-2}$ | $1.78e^{-1}$ |
| | 0.1 | 0.5 | $3.26e^{-2}$ | $9.57e^{-3}$ | $9.94e^{-1}$ | $4.11e^{-4}$ | $2.30e^{-3}$ | $9.48e^{-1}$ |
| | 0.3 | 0.05 | $\mathbf{2.02e^{-2}}$ | $4.23e^{-2}$ | $5.94e^{-1}$ | $2.55e^{-2}$ | $7.45e^{-2}$ | $\mathbf{1.27e^{-1}}$ |
| | 0.3 | 0.1 | $1.52e^{-2}$ | $3.94e^{-2}$ | $1.80e^{-1}$ | $\mathbf{5.96.e^{-5}}$ | $\mathbf{2.94.e^{-4}}$ | $8.10e^{-1}$ |
| | 0.3 | 0.5 | $1.37e^{-1}$ | $3.14e^{-1}$ | $8.15e^{-1}$ | $4.05e^{-4}$ | $3.53e^{-3}$ | $2.97e^{-1}$ |

## E.2    Evaluation under Varying Number of Clients

Here, we describe how the performance of DP-FedAvg and AdaptDP-FedAvg changes with varying numbers of clients. We empirically evaluated the training loss for each dataset using $P = 2, 4, 6$ clients. For each dataset, we set the number of training samples per client to $n_{\text{train}} = 3000, 6000, 9000$ for $P = 2, 4, 6$ clients, respectively. The results are presented in Table 4. As shown in Table 4, although the training loss varies depending on the number of clients, AdaptDP-FedAvg generally outperforms DP-FedAvg.

Table 4: Training loss for each hyperparameter setting tested.

| Dataset | Num of client | DP-FedAvg | | | AdaptDP-FedAvg | | |
|---------|---------------|-----------|------|------|----------------|------|------|
| | | Lv.1 | Lv.2 | Lv.3 | Lv.1 | Lv.2 | Lv.3 |
| T1) Artificial dataset $(R,K)=(150,20)$ | 2 | $6.73e^{-6}$ | $2.69e^{-5}$ | $7.62e^{-5}$ | $\mathbf{3.34e^{-7}}$ | $\mathbf{3.81e^{-6}}$ | $\mathbf{1.82e^{-5}}$ |
| | 4 | $4.44e^{-6}$ | $1.78e^{-5}$ | $4.93e^{-5}$ | $\mathbf{2.00e^{-7}}$ | $\mathbf{2.49e^{-6}}$ | $\mathbf{1.23e^{-5}}$ |
| | 6 | $3.05e^{-6}$ | $1.22e^{-5}$ | $3.38e^{-5}$ | $\mathbf{1.39e^{-7}}$ | $\mathbf{1.72e^{-6}}$ | $\mathbf{7.76e^{-6}}$ |
| T3) MNIST $(R,K)=(100,15)$ | 2 | $4.84e^{-5}$ | $5.57e^{-5}$ | $4.06e^{-4}$ | $\mathbf{4.25e^{-5}}$ | $\mathbf{3.38e^{-5}}$ | $\mathbf{4.02e^{-5}}$ |
| | 4 | $3.79e^{-5}$ | $4.28e^{-5}$ | $7.53e^{-5}$ | $\mathbf{3.09e^{-5}}$ | $\mathbf{3.38e^{-5}}$ | $\mathbf{4.11e^{-5}}$ |
| | 6 | $\mathbf{4.28e^{-4}}$ | $1.99e^{-3}$ | $\mathbf{4.29e^{-3}}$ | $1.19e^{-3}$ | $1.60e^{-3}$ | $4.51e^{-3}$ |
| T4) FashionMNIST $(R,K)=(100,15)$ | 2 | $2.02e^{-2}$ | $3.39e^{-2}$ | $1.28e^{-1}$ | $\mathbf{5.96e^{-5}}$ | $\mathbf{2.94e^{-4}}$ | $\mathbf{1.27e^{-1}}$ |
| | 4 | $4.96e^{-5}$ | $5.75e^{-2}$ | $5.98e^{-2}$ | $\mathbf{4.92e^{-5}}$ | $5.75e^{-2}$ | $\mathbf{5.86e^{-2}}$ |
| | 6 | $4.04e^{-5}$ | $5.55e^{-2}$ | $1.09e^{-1}$ | $\mathbf{3.88e^{-5}}$ | $\mathbf{5.52e^{-2}}$ | $\mathbf{1.08e^{-1}}$ |

## E.3 Evaluation using a non-convex model

Although AdaptDP-FedAvg is theoretically guaranteed only for convex models, we experimentally evaluated its performance compared to DP-FedAvg on the image classification task (T3: MNIST) with $P = 2$ using a non-convex model. Specifically, we employed MobileNet as the model architecture, with $\eta = 0.3$ and $C, \hat{G} = 0.05$. As shown in Table 5, even when using a non-convex model, AdaptDP-FedAvg achieves lower training loss compared to DP-FedAvg. While this observation lacks theoretical guarantees, it aligns with our expectation that the proposed method remains effective for overparameterized deep neural networks, even in the non-convex setting.

Table 5: Training loss using non-convex model.

| Dataset | DP-FedAvg | | | AdaptDP-FedAvg | | |
|---------|-----------|------|------|----------------|------|------|
| | Lv.1 | Lv.2 | Lv.3 | Lv.1 | Lv.2 | Lv.3 |
| T3) MNIST $(R,K)=(100,15)$ | $\mathbf{8.12e^{-5}}$ | $1.24e^{-4}$ | $4.40e^{-5}$ | $1.35e^{-4}$ | $\mathbf{9.13e^{-5}}$ | $\mathbf{2.69e^{-5}}$ |

## E.4 Evaluation additional computational costs

As noted in Remark 1, AdaptDP-FedAvg requires additional computational costs compared to DP-FedAvg. We conducted additional experiments on the image classification task (T3: MNIST) with $P = 2$ clients to measure the runtime per communication round as well as the peak memory usage. As shown in Table 6, we experimentally found that both the runtime and peak memory usage of AdaptDP-FedAvg and DP-FedAvg are nearly identical, indicating that the additional cost introduced by AdaptDP-FedAvg is practically negligible.

Table 6: the runtime per communication round and the peak memory usage.

| Dataset | DP-FedAvg | | AdaptDP-FedAvg | |
|---------|-----------|---|----------------|---|
| | runtime[Sec] | CPU/GPU memory usage[GB] | runtime[Sec] | CPU/GPU memory usage[GB] |
| T3) MNIST $(R,K)=(100,15)$ | 19.60 | 2.10/1.62 | 19.63 | 2.10/1.62 |

## F Impact statement

We propose a new adaptive clipping algorithm for local record-level DP federated learning. While our method provides theoretical guarantees of statistical data privacy under differential privacy, it does not ensure complete immunity to data leakage. Consequently, there remains a potential risk of privacy breaches.

