# OpenReview forum: "Adaptive Clipping for Differential Private Federated Learning in Interpolation Regimes"
_TMLR — Accepted by TMLR_

### Review · Reviewer_Ed7n · 2025-04-21

**Summary Of Contributions:**

This paper proposes a new adaptive clipping algorithm that adaptively sets the clipping threshold based on the gradient norms of the current model in a federated learning regime. The proposed algorithm provides differential privacy guarantees and achieves a better utility bound than the classical DP-SGD in theory. The experimental results support the theoretical findings.

**Audience:**

Yes

**Broader Impact Concerns:**

There are no concerns on the ethical implications of this work.

**Claims And Evidence:**

Yes

**Requested Changes:**

My detailed requested changes are listed in the Weakness part; please refer to it.

**Strengths And Weaknesses:**

Strengths:

1. The investigated differentially private federated learning problem is of paramount importance in practice.

2. The proposed algorithm is novel and achieves a better theoretical bound.

Weaknesses:

1. Assumption 2 (the interpolation regime assumption) restricts the application scenario of the proposed algorithm to overparameterized models and fails to generalize to more practical settings.

2. In the numerical experiments, the authors consider only a small-scale federated regime with two clients. I suggest that the authors evaluate the proposed algorithm with a larger number of clients to demonstrate its general performance, and further investigate the impact of the number of clients on the algorithm's behavior by varying the client size in the experiment.

---

> ### Author Response · Authors · 2025-05-23
>
> > Assumption 2 (the interpolation regime assumption) restricts the application scenario of the proposed algorithm to overparameterized models and fails to generalize to more practical settings.
>
> We would like to clarify that Assumption 2 (the interpolation regime assumption) does not assume the model fits the true data distribution, but rather assumes interpolation of the training data. This is known to naturally occur in overparameterized settings, such as approximate kernel methods and overparameterized deep neural networks. Moreover, this assumption is widely adopted in the recent papers, and is considered a standard and practical assumption in the context of modern machine learning. Modern neural networks used for training are typically large and overparameterized, and we believe that the interpolation assumption is satisfied realistic and well-justified even in more practical settings. Consequently, our proposed algorithm is expected to be applicable and effective in such settings as well.
>
> >In the numerical experiments, the authors consider only a small-scale federated regime with two clients.
>
> Thank you for your suggestion. To reflect your comment, we conducted additional experiments with $P = 4, 6$ clients to evaluate the scalability of adaptDP-FedAvg. The results demonstrated that our method consistently outperforms DP-FedAvg even as the number of clients increases. We have added these results to the appendix E.2.

---

### Review · Reviewer_dzJ3 · 2025-05-02

**Summary Of Contributions:**

This paper introduces AdaptDP-FedAvg, a method to improve privacy-preserving federated learning by dynamically adjusting the "clipping radius" (a parameter that limits gradient sizes to protect privacy). The main contributions include:
1. Adaptive Clipping: Instead of using a fixed clipping radius (common in prior methods like DP-SGD), the new algorithm adjusts it based on the average size of gradients. This reduces unnecessary noise, improving model accuracy while maintaining privacy.
2. Theoretical Guarantees: The author proves the method meets differential privacy standards (using Rényi Differential Privacy) and shows it achieves better "utility" (model performance) than fixed-radius methods for smooth and non-strongly convex problems. The improvement is rooted in the "interpolation property," where gradients shrink as the model nears the optimal solution.
3. Experiments show the effectiveness over the baseline method (DP-FedAvg).

**Audience:**

Yes

**Broader Impact Concerns:**

NA.

**Claims And Evidence:**

Yes

**Requested Changes:**

Relax Assumptions:
Consider cases where objectives are non-convex.
Simplify Tuning:
Provide guidance or an automated way to set the new hyperparameter (τ).

**Strengths And Weaknesses:**

Strengths:
1. Smart Noise Reduction: By adapting the clipping radius to actual gradients, the method reduces noise without sacrificing privacy, a key improvement over one-size-fits-all approaches.
2. Solid Theory: The paper provides a theoretical analysis linking the adaptive strategy to better utility.
3. Practical Use: Addresses a real need in federated learning, where balancing privacy and model accuracy is critical (e.g., in healthcare or finance data).

Weaknesses:
1. Assumptions: The theoretical benefits rely on convexity, which may not hold for all real-world models (e.g., deep neural networks with non-convex losses).
2. Hyperparameter Tune: Introduces a new parameter (τ) that needs manual tuning.

---

> ### Author Response · Authors · 2025-05-23
>
> > Relax Assumptions: Consider cases where objectives are non-convex.
>
> Thanks for your feedback. While we appreciate your suggestion, the theoretical extension to the non-convex setting is left as future work, as it lies beyond the main scope of this paper. Instead, we have conducted additional experiments using a non-convex model on the MNIST image classification task. As demonstrated in Appendix E.3, the results demonstrate that AdaptDP-FedAvg still outperforms DP-FedAvg, similarly to the convex setting.
>
> >Simplify Tuning: Provide guidance or an automated way to set the new hyperparameter (τ).
>
> Through our experiments, we obtained empirical insights indicating that $\tau = 1$ tends to yield good performance for the models and datasets we used. However, the optimal $\tau$ can vary depending on the specific model and data, and at present, there is no principled method to determine a universally optimal $\tau$. In more complex experiment settings—such as when some per-sample gradients do not become small—setting a larger $\tau$ may be beneficial for ensuring effective adaptive clipping. Nonetheless, selecting an optimal $\tau$ still requires a greedy search process, similar to tuning the learning rate. We consider this an important direction for future work. We have added this discussion to Section 5.2.

---

### Review · Reviewer_1V8F · 2025-05-10

**Summary Of Contributions:**

Adaptive Clipping Algorithm: The authors propose AdaptDP-FedAvg, an adaptive gradient clipping method for differentially private (DP) federated learning (FL). The clipping radius is dynamically adjusted based on the root-mean-square of gradient norms, exploiting the interpolation property (where per-sample gradients vanish near optimality).

Theoretical Analysis: They provide (ε, δ)-DP guarantees via Rényi DP (RDP) and utility analysis for smooth, non-strongly convex objectives, showing improved convergence rates over DP-SGD under interpolation.

Empirical Validation: Experiments on synthetic and real-world datasets (MNIST, FashionMNIST, Bank Marketing) demonstrate that AdaptDP-FedAvg achieves lower training loss compared to DP-FedAvg with fixed clipping, especially in high-privacy regimes.

**Audience:**

Yes

**Broader Impact Concerns:**

I do not find any concerns of broader impact

**Claims And Evidence:**

Yes

**Requested Changes:**

Essential Revisions:

Generalization Analysis: Discuss why lower training loss sometimes correlates with reduced test accuracy (e.g., MNIST). Explore regularization techniques or early stopping to mitigate overfitting.

Hyperparameter Guidance: Provide practical guidelines for selecting τ and Ĝ, possibly via sensitivity analysis or dataset-specific heuristics.

Comparison with State-of-the-Art: Include experiments/baselines against other adaptive clipping methods (e.g., Asi et al.’s PAGAN, Clipless DP-SGD) to contextualize improvements.

Computational Overhead: Quantify the runtime/memory impact of adaptive clipping (e.g., per-round time vs. DP-FedAvg).

Recommended Improvements:

Theoretical Limitations: Clarify assumptions (e.g., interpolation) and their real-world applicability. Discuss how non-interpolation settings might affect performance.

Broader Impact Statement: Address ethical considerations (e.g., fairness implications of adaptive clipping in heterogeneous FL).

Visualization: Include plots showing the relationship between τ_q and utility improvement to reinforce theoretical claims.

Optional Enhancements:

Code Release: Publish implementation to ensure reproducibility.

Large-Scale Experiments: Test on larger models/datasets (e.g., CIFAR-10, ResNet) to assess scalability.

**Strengths And Weaknesses:**

Strengths:

Problem Relevance: Addresses a critical challenge in FL—balancing privacy and utility—by reducing clipping bias and wasted privacy budget.

Theoretical Rigor: Combines RDP-based privacy analysis with utility guarantees, rigorously linking adaptive clipping to improved convergence rates in interpolation regimes.

Empirical Support: Demonstrates consistent improvements in training loss across tasks, validating the theoretical claims.

Practicality: Highlights negligible additional communication/computation costs (scalar-valued clipping radius updates).

Weaknesses:

Generalization Concerns: While training loss improves, test accuracy occasionally underperforms DP-FedAvg (e.g., MNIST), suggesting potential overfitting. The paper attributes this to training loss focus but lacks deeper analysis.

Hyperparameter Sensitivity: Introduces new hyperparameters (τ, Ĝ) with limited guidance on tuning. The theoretical benefits depend on dataset-dependent quantities like τ_q, raising questions about robustness.

Limited Baseline Comparison: Focuses on DP-FedAvg; comparisons with recent adaptive clipping methods (e.g., PAGAN, Clipless DP-SGD) are missing.

Computational Overhead: While claimed as negligible, no empirical profiling of runtime/memory costs is provided.

---

> ### Author Response · Authors · 2025-05-23
>
> >Generalization Analysis:
>
> Thank you for your insightful comment.  In Table 2 (T3) for MNIST Test Accuracy, while the average performance of DP-FedAvg exceeds that of AdaptDP-FedAvg, it should be noted that the variance for DP-FedAvg is very large (e.g., $5.5$ and $7.25$), whereas the variance of AdaptDP-FedAvg remains small.  This suggests that although DP-FedAvg may appear to perform better on average, this is likely due to a few seeds that happened to produce exceptionally good results. It does not indicate that DP-FedAvg consistently outperform AdaptDP-FedAvg. We consider that evaluating the highest test accuracy reported in Table 2 is, in practice, similar to tuning each method and employing early stopping. Although we did not employ any regularization techniques in this work, we acknowledge that various regularization-based approaches exist to reduce variance. Analyzing and experimentally evaluating integration of such techniques into our proposed method is an important direction for future work.
>
> >Hyperparameter Guidance:
>
> Through our experiments, we obtained empirical insights indicating that $\tau = 1$ and Approximately $\hat{G} = 0.05$ tends to yield good performance for the models and datasets we used. However, the optimal $\tau, \hat{G}$ vary depending on the models and datasets, and there is currently no principled way to determine them universally. Similar to learning rate selection, a greedy search remains necessary. We regard this as an important direction for future work. While this introduces the additional overhead of tuning additional hyperparameters, a main contribution of this paper is the design of an algorithm that, under a reasonable assumption, allows for adaptive clipping while still enabling successful training—with theoretical convergence guarantees. We have added this discussion to Section 5.2.
>
> >Comparison with State-of-the-Art: Include experiments/baselines against other adaptive clipping methods (e.g., Asi et al.’s PAGAN, Clipless DP-SGD) to contextualize improvements.
>
> We appreciate your suggestion, and thank you for pointing out relevant adaptive clipping methods. However, we did not conduct experiments comparing our method with these two approaches, as such a comparison presents several challenges, outlined below:
>
> PAGAN is not designed for federated learning, and adapting it to an FL setting would require developing a new algorithm adapted to FL as well as conducting differential privacy theoretical analysis, which is nontrivial. Therefore, This makes it difficult to ensure a fair comparison with AdaptDP-FedAvg.
> Furthermore, the idea of PAGAN lies in adapting the noise scale along each coordinate direction, which is fundamentally orthogonal to our approach that performs adaptation at the iteration level.
>
> Moreover, Clipless DP-SGD is designed for Lipschitz networks, as specified in Definition 3 of Section 1 in their paper, whereas AdaptDP-FedAvg targets general neural networks. Therefore, applying Clipless DP-SGD in our experiment setting would be inappropriate and would not result in a fair comparison.
>
> >Computational Overhead:
>
> As noted in Remark 1, AdaptDP-FedAvg introduces additional costs compared to DP-FedAvg—namely, the computational cost of evaluating a scalar $C_r$ per round and a communication cost of $\Theta(P)$ instead of $\Theta(Pd)$. Furthermore, to empirically support this analytical observation, we conducted additional experiments in the image classification task (T3: MNIST) to measure the runtime per communication round as well as the peak memory usage. The results showed that AdaptDP-FedAvg required $19.63$ seconds, while DP-FedAvg required $19.60$ seconds, indicating that the difference in runtime is practically negligible. Furthermore, the peak memory usage was identical for both methods: 2.1 GB of CPU memory and 1.62 GB of GPU memory. we appreciate your insightful comment. We have added this information in the appendix E.4.
>
> >Theoretical Limitations:
>
> As stated just below Assumption 2, it is known that interpolation regimes commonly can be satisfied when training high-dimensional models with universality properties, such as kernel methods and modern deep neural networks. In our study, we selected model sizes for which the interpolation regime was satisfied, as indicated by preliminary experiments. It is also well known that neural networks commonly used in modern machine learning practice are typically large and overparameterized.  We believe this assumption will not significantly restrict real-world applicability.
>
> On the other hand, when the interpolation regimes cannot be satisfied, our algorithm is unlikely to outperform DP-FedAvg in terms of utility. This is because the norms of the loss gradients fail to converge to zero even near the optimal solution, which prevents both the clipping radius and the magnitude of DP noise from decreasing.
>
> >Broader Impact Statement:
>
> We have added an impact statement in the appendix F.

---

> > ### Author Response · Authors · 2025-05-23
> >
> > > Visualization:
> >
> > Thank you for your suggestion. However, it is difficult to implement for the following reasons:
> >
> > First of all, please note that $\tau_{q}$ is not a hyperparameter that can be arbitrarily set by us. For this reason, we can neither experimentally nor theoretically investigate the dependence of performance by varying $\tau_q$.
> > While it might be possible to observe a relationship between $\tau_q$ and utility improvement by measuring the estimated $\tau_q$ and corresponding utility improvement for each dataset, it is difficult to isolate and fairly evaluate the specific impact of $\tau_q$, as training performance generally depends on many other factors beyond $\tau_q$.
> >
> > >Large-Scale Experiments:
> >
> > To reflect your comments, we conducted additional experiments on the MNIST image classification task using a non-convex model (MobileNet). The results demonstrate that AdaptDP-FedAvg consistently outperforms DP-FedAvg, similarly to the convex setting. We have added these results to the appendix E.3.

---

### Decision · Action_Editor_mAb5 · 2025-06-09

**Recommendation:** Accept as is

**Audience:**

Yes

**Audience Explanation:**

The paper proposes an adaptive gradient clipping method for differentially private federated learning. Privacy is an important issue in federated learning, which has many applications in the big data era. The adaptive clipping scheme allows a better trade-off between privacy and utility as compared to the fixed-clipping method. The proposed method should be interesting in both the differentially private machine learning community and the optimization community.

**Claims And Evidence:**

Yes

**Claims Explanation:**

The paper conducts rigorous theoretical analysis to show that the proposed method enjoys privacy guarantees and convergence rates. The paper shows the proposed method enjoys improved convergence rates under interpolation setting. All the reviewers appreciate the soundness of the theoretical analysis. The paper also conducts comprehensive empirical analysis on various datasets to show that  the proposed method achieves lower training loss compared to existing methods, especially in high-privacy regimes.